# Ferroelectric and Non-Linear Optical Nanofibers by Electrospinning: From Inorganics to Molecular Crystals

**DOI:** 10.3390/nano15050409

**Published:** 2025-03-06

**Authors:** Rosa M. F. Baptista, Etelvina de Matos Gomes, Michael Belsley, Bernardo Almeida

**Affiliations:** Centre of Physics of Minho and Porto Universities (CF-UM-UP), Laboratory for Physics of Materials and Emergent Technologies (LaPMET), University of Minho, Campus de Gualtar, 4710-057 Braga, Portugal; emg@fisica.uminho.pt (E.d.M.G.); belsley@fisica.uminho.pt (M.B.)

**Keywords:** piezoelectric ceramics, molecular crystals, electrospinning, nanofibers and microfibers, ferroelectrics, multiferroics, pyroelectricity, piezoelectricity, nonlinear optics, hybrid functional materials, energy harvesting

## Abstract

In recent decades, substantial progress has been made in embedding molecules, nanocrystals, and nanograins into nanofibers, resulting in a new class of hybrid functional materials with exceptional physical properties. Among these materials, functional nanofibers exhibiting ferroelectric, piezoelectric, pyroelectric, multiferroic, and nonlinear optical characteristics have attracted considerable attention and undergone substantial improvements. This review critically examines these developments, focusing on strategies for incorporating diverse compounds into nanofibers and their impact on enhancing their physical properties, particularly ferroelectric behavior and nonlinear optical conversion. These developments have transformative potential across electronics, photonics, biomaterials, and energy harvesting. By synthesizing recent advancements in the design and application of nanofiber-embedded materials, this review seeks to highlight their potential impact on scientific research, technological innovation, and the development of next-generation devices.

## 1. Introduction

Nanofibers have emerged as a promising platform for advancing optical and electrical applications due to their unique structural and functional properties. Traditional piezoelectric ceramics, such as lead zirconate titanate (PZT), zinc oxide (ZnO), and barium titanate (BaTiO_3_), exhibit remarkable piezoelectric performance. However, their inherent brittleness and the need for high processing temperatures (T > 600 °C) significantly constrain their practical utility [1]. These limitations have driven the search for novel ferroelectric materials that are both flexible and more easily processed. Among these alternatives, ferroelectric fiber composites fabricated by electrospinning hold significant promise. These materials not only exhibit enhanced ferroelectric, piezoelectric, pyroelectric, and non-linear optical (NLO) properties, but are also flexible, biocompatible, and environmentally friendly.

Ferroelectricity is a characteristic of crystals with spontaneous polarization, which can alternate between two or more equivalent orientational states under an external electric field. These materials, belonging to a subset of pyroelectric crystals, inherently exhibit piezoelectricity [2]. Importantly, many practical applications of ferroelectric materials leverage their related properties, such as pyroelectricity and piezoelectricity, rather than ferroelectricity itself. For instance, pyroelectric materials find utility in imaging and detection applications [3,4], while piezoelectric materials are widely used in electromechanical devices. Notably, the high piezoelectric coefficients of materials such as PZT have enabled advancements in microelectromechanical systems (MEMSs) and energy harvesting devices, including their miniaturization [3,5,6,7,8]. Ferroelectric materials are also widely used in capacitors for microelectronics, non-volatile memories, and sensors [9,10,11,12].

A key advantage of reducing ferroelectric materials to nanoscale dimensions is the substantial decrease in voltage required to achieve the coercive fields of typical ferroelectrics (about 50 kV/cm), making them highly attractive for integration into modern electronic systems. For example, nanoscale ferroelectric structures require potential differences as low as 5 V to achieve these same coercive fields, compared to several kV in bulk crystals or ceramics [9].

Nanostructures with reduced dimensions, in the form of fibers, wires, rods, belts, tubes, and rings, have attracted significant interest due to their unique properties and potential applications in sensing, flexible electronics, and nanoscale electrical energy generation. Among these, ferroelectric nanofibers offer unique advantages, as they do not require substrates and enable new functionalities. While several fabrication methods, including chemical vapor deposition (CVD) [13,14], hydrothermal methods [15], solid-state reactions [16,17], and sol–gel [18,19,20], have been used to create nanostructures, these methods often involve complex procedures, expensive equipment, or costly precursor materials.

In contrast, electrospinning is a simple, versatile and cost-effective technique capable of producing continuous, individual fibers with a variety of structures, including composite nanofibers with inclusions, core–shell, and 3D architectures [21,22,23,24,25]. Electrospinning enables the fabrications of nanofibers with diameters ranging from a few micrometers to less than 100 nm. Electrospinning stands out as a straightforward and adaptable technique capable of creating nanofibers from a wide range of polymers, often incorporating organic or semi-organic nanocrystals to create hybrid composite materials. This method has also been used to align nonlinear optical (NLO) chromophores in subwavelength structures with tailored functionalities [26].

Low-dimensional nanostructures—specifically, ferroelectric nanofibers—have been successfully fabricated using the versatile electrospinning technique [27,28,29,30,31,32]. These nanofibers exhibit exceptional ferroelectric properties at the nanoscale, making them highly promising for applications ranging from flexible electronic devices and highly sensitive sensors, to nanogenerators for power generation. Additionally, electrospinning facilitates the integration of multiple functionalities in a single system, such as combining ferroelectric and magnetic properties within a single nanostructured entity. The demand for cost-effective sensors, actuators, and electronic components, coupled with the need for increased device integration, has spurred interest in exploring nanofibers with high aspect ratios and multifunctionalities. Electrospun nanofibers offer an optimal solution, combining affordability, flexibility, and versatility.

Electrospun polymer fibers embedded with organic molecules have gained significant attention due to their versatility and wide-ranging applications in biomedicine and electronics. In the biomedical domain, these fibers are extensively used for tissue-engineering scaffolds and controlled drug-delivery systems, as they mimic the extracellular matrix (ECM) and provide a high surface-area-to-volume ratio conducive to cell adhesion, proliferation, and differentiation [33]. Additionally, it is possible to endow electrospun nanofibers with diverse physical or chemical properties by incorporating specific tailored nanoparticles [34,35,36].

In photonics, electrospun fibers exhibit promising nonlinear optical properties and enhanced light emission characteristics. As will be explored in this review, notable second harmonic generation (SHG) has been observed in several polymer-organic and semi-organic electrospun systems, underscoring their potential for frequency-doubling applications. Furthermore, conjugated polymer fibers have demonstrated enhanced polarized photoemission coupled with self-waveguiding behavior [37,38], which is particularly advantageous for nanophotonic devices and sensors in the blue spectral region, an area where solid-state photonic sources are challenging to develop. These optical functionalities are attributed to the alignment of the molecular dipoles by the strong electric fields employed in the electrospinning process and the smooth morphology and flexibility of the fibers, which support efficient waveguiding and light manipulation.

Incorporating ferroelectric organic and semi-organic nanocrystals into polymeric fibers through electrospinning has led to the development of sensors and energy harvesting systems. This approach can potentially harness both the pyroelectric and piezoelectric effects, synergistically combining the functional properties of nanocrystals with the flexibility and high surface area of electrospun fibers. Applications of these nanofiber-based ferroelectric systems span a broad spectrum of applications, with particular emphasis on energy-harvesting technologies. Several recent reviews focus on the applications of ferroelectric composites and piezoelectric electrospun fibers [11,39,40,41,42,43,44,45,46,47]. However there remains a notable gap in the literature concerning the fundamental physical properties of electrospun single-phase inorganic ferroelectric nanofibers and organic ferroelectric nanofibers beyond polyvinylidene fluoride (PVDF).

In summary, while the synthesis and applications of nanofibers have been extensively studied, their ferroelectric and nonlinear optical properties are often overlooked. This review aims to address this gap by focusing on the ferroelectric and nonlinear optical characteristics of electrospun nanofibers, emphasizing the unique features and potential applications of both inorganic and organic ferroelectrics.

## 2. Electrospinning Process and Parameters

### 2.1. Electrospinning

Electrospinning is a widely used technique for synthesizing nanofibers based on the stretching of a viscoelastic solution and its consequent solidification under the influence of a strong electric field. Figure 1a illustrates the scheme of a typical electrospinning system used for producing fiber mats. In this method, a high-DC electric field (100–500 kV/m) is applied to a metallic capillary containing a polymer or sol–gel precursor solution. At the tip of the capillary, an electrically charged drop forms, with electric charges evenly distributed across its surface (Figure 1b).

The interplay between the externally applied electric field, repulsive forces among surface charges, and the surface tension of the droplet induces a characteristic cone-shaped deformation known as the Taylor cone, typically exhibiting an angle of approximately 30° at its tip [22]. When the applied electric field exceeds a critical threshold, the electrostatic forces overcome the surface tension, leading to the ejection of a fine jet from the tip of the Taylor cone. The critical potential difference $V_{c}$ required for jet emission, assuming a spherical shape of the droplet at the cone’s apex, is given by [25]:(1)$$V_{c}^{2}=0.36{\left(\frac{D}{L}\right)}^{2}\left(ln\left(\frac{2L}{R}\right)-\frac{3}{2}\right)\left(1.3\pi R\gamma \right).$$

Here, *D* is the distance over which the high voltage is applied, *R* is the radius of the syringe needle, *L* is the length of the fluid column that experiences the accelerating electric field, and *γ* is the surface tension of the solution. All distances are measured in cm, with the surface tension expressed in dyn/cm, yielding a critical tension, $V_{C}$ in kV.

Once the jet is emitted, it undergoes stretching due to the applied electric field, accompanied by instabilities arising from charge interactions. These instabilities result in the elongation and thinning of the jet, reducing its diameter as the solvent evaporates. In this way, fibers are obtained that can reach diameters down to a few tens of nanometers and lengths of several tens of meters (Figure 1c), in a controlled manner.

By tailoring the composition of the precursor solutions, it is possible to incorporate functional materials, such as ferroelectric or magnetic nanoparticles, into the fibers. This enables the creation of functionalized composite nanofibers with tunable properties for targeted applications. The high surface-to-volume ratio and aspect ratio (length to diameter) of the resulting fibers make electrospinning a versatile and powerful technique for fabricating micro- and nanoscale fibers with specific functionalities.

The final fiber diameter ($d_{f}$) is determined by the balance between surface tension and the repulsion of surface charges in the stretched jet. At the point where the jet ceases to thin further, the fiber diameter can be expressed as [25]:(2)$$d_{f}={\left(\gamma \varepsilon^{\prime }{\left(\frac{Q}{I}\right)}^{2}\frac{2}{\pi \left(2ln\left(\chi \right)-3\right)}\right)}^{1/3}$$
where *Q* is the flow rate of the fluid jet, *I* is the electric current through the fluid, *ε′* is the permittivity of the environment (air), *γ* is the fluid surface tension, and *χ* is the dimensionless wavelength of the jet’s torsional instability (proportional to the curvature radius of the bending perturbation divided by the jet radius). Consequently, the fiber diameter depends not only on the physical properties of the precursor solution (e.g., concentration, viscosity, conductivity) but also on process parameters such as the applied electric field, flow rate, and collection distance. Fiber diameter uniformity can be monitored by observing the stability of the electric current during the process. Any fluctuations in current often indicate variations in jet stability.

### 2.2. Process Parameters

The properties of the precursor solution and the corresponding processing parameters significantly influence the morphology and characteristics of the electrospun fibers. Key factors include polymer concentration, solvent properties, applied voltage, needle-to-collector distance, flow rate, and environmental conditions.

As polymer concentration or viscosity increases, the average fiber diameter tends to grow. This occurs because the higher viscosity resists elongation, slowing the jet’s thinning process. Conversely, low-viscosity solutions often result in thinner fibers or, if the concentration is too low, the formation of droplets instead of continuous fibers (a phenomenon known as electrospraying).

Solvents with low vapor pressure typically produce thinner fibers. This is because slower solvent evaporation prolongs the jet’s elongation and diameter reduction phases before solidification occurs. The choice of solvent also influences the solution’s conductivity and surface tension, both of which directly affect fiber formation. Higher solution conductivity increases the electric charge carried by the jet, enhancing the electric elongation force. As a result, the fiber diameter decreases due to increased stretching. Equation (2) illustrates this relationship, where the electric current (I) is proportional to the charge density.

The applied electric field, determined by the voltage and needle-to-collector distance, must exceed a threshold to initiate jet emission (Equation (1)). For larger electric fields, the fiber diameter tends to increase slightly because more fluid is ejected. However, this effect is less pronounced compared to the influence of solution viscosity.

A shorter needle-to-collector distance reduces the jet’s elongation time, resulting in thicker fibers. Conversely, increasing the distance allows more time for jet stretching and solvent evaporation, leading to thinner fibers. On the other hand, excessive distances can weaken the electric field, reducing the stability of fiber formation.

Reducing the needle radius or decreasing the flow rate results in thinner fibers. Smaller needle radii create finer initial jets, while lower flow rates reduce the amount of fluid ejected, enhancing the elongation effect.

In addition to these parameters, the technique is also affected by the environmental conditions, such as the relative humidity and ambient temperature. Higher humidity generally leads to thinner fibers because slower drying allows for extended stretching of the jet. Conversely, excessive humidity can prevent proper drying, causing the jet to break into droplets and inhibiting fiber formation. Similarly, temperature affects solvent evaporation rates, with higher temperatures facilitating faster solidification.

Electrospinning is highly adaptable, enabling the creation of nanofibers with intricate structures by varying needle and collector configurations. In coaxial electrospinning, two needles are concentrically aligned, such that one precursor solution can be delivered through an inner capillary while the other can pass through the outer capillary, creating tubular or core–shell fibers. With side-by-side needles, bi- or three-phase fibers can be obtained [23,48]. Aligned fibers can be obtained using rotating drums or parallel threads, which is advantageous for applications based on fiber orientation [21,49,50]. Using more complex electrodes, such as metallic wires coupled to mobile positioners, liquid collectors, and other assemblies [23,51,52], three-dimensional structures can be obtained.

## 3. Composite Ferroelectric and Multiferroic Fibers with Inorganic Inclusions

Recent studies on ferroelectric nanowires have revealed interesting size-dependent properties that deviate significantly from their bulk counterparts. These nanostructures not only retain ferroelectric properties at extremely small diameters but can also exhibit enhanced axial polarization with decreasing size [53] and significant ferroelectric and piezoelectric responses [54]. This underscores the potential for novel ferroelectric applications of functionalized nanofibers.

X-ray diffraction measurements on BaTiO_3_, PbTiO_3_, and PZT ferroelectric nanowires have confirmed the preservation of their polar structure. However, these nanostructures often display reduced spontaneous ferroelectric deformation and tetragonality compared to macroscopic crystals [55]. A key finding is that below the Curie temperature, spontaneous polarization develops parallel to the wire’s axis, with the [001] direction of the tetragonal structure aligning along this axis [49,56,57]. This alignment has significant implications for the anisotropic properties of these nanowires.

Traditionally, the fabrication of ferroelectric nanowires has relied on complex and costly techniques. In this context, electrospinning emerges as a promising, cost-effective alternative for producing functionalized ferroelectric nanofibers. This technique offers advantages in terms of process control and versatility, enabling the synthesis of a wide range of ferroelectric and multiferroic nanofibers with tailored properties.

This section focuses on the synthesis of composite electrospun nanofibers based on ferroelectric and multiferroic materials. The discussion of ferroelectrics and multiferroics is linked to the deep relationship between them. In particular, they share the requirement for good electrical insulation. This characteristic is crucial whether considering intrinsic multiferroics or composite systems involving multiple materials. By exploring these relationships, we aim to elucidate the potential of electrospinning in creating multifunctional nanofibers with enhanced ferroelectric and multiferroic properties.

### 3.1. Ferroelectric Polymers

Ferroelectric polymers encompass a wide range of materials, including poly-L-lactic acid (PLLA), polylactic glycolic acid (PLGA), and, most notably, poly(vinylidene fluoride) (PVDF) and its copolymers [50,58,59,60,61]. Among these, PVDF and its derivatives are the best known and most widely used.

PVDF, particularly in its β phase, exhibits strong ferroelectric properties with a high electromechanical response. The electrospinning technique is especially effective for preparing PVDF nanofibers. Depending on the precursor solution PVDF concentration, nanofibers with diameters in the range of 60–400 nm can be prepared [50,58,59,60,61]. Notably, electrospinning promotes the formation of the β (polar) phase in PVDF nanofibers, especially in thinner fibers [58,61], due to mechanical elongation and polarization induced by the applied electric field. In PVDF nanofibers, the c-axis of crystallites tends to align along the fiber axis, enhancing their piezoelectric properties.

Both natural and synthetic polyamides can display ferroelectric properties [62,63,64,65], with nylons being the most extensively researched synthetic ferroelectric polyamides [66]. The ferroelectric hysteresis characteristics identified in even-numbered nylons were initially documented in nylon-6 subjected to quenching from the melt, attributed to the autonomous rotation of dipoles within the structurally unstable nematic-type configuration [67]. The molecular mobility of dipolar entities within polyamide 6,9 was systematically investigated using a synergistic approach that combined thermo-stimulated current (TSC) and dynamic dielectric spectroscopy (DDS). Significantly, this study marked a pioneering effort in quantifying polarization and piezo-/pyroelectric coefficients, thereby elucidating the distinctive ferroelectric behavior inherent in polyamide-6,9. Ferroelectric activity was detected in nylon-6,9 subsequent to polarization up to 100 MV m^−1^ [68].

Odd-numbered nylons, including nylon-77, nylon-79, nylon-97, and nylon-99, display ferroelectric behavior due to the parallel arrangement of amide dipoles within crystalline regions. This phenomenon mirrors the characteristics observed in nylon comprised of ω-aminocarboxylic acid with an odd number of carbon atoms [69]. Nylon-11, a key odd-numbered nylon, shows promise as a functional material for electronic devices or energy storage [70,71,72]. It is crucial to control the formation of metastable crystalline phases, such as γ-piezoelectric or δ′-ferroelectric, for these applications. A recent method was described for manufacturing metastable γ-phase nylon-11 fibers using electrospinning with a customized solvent system [73].

Liquid crystal polymers represent a distinct subclass of ferroelectric polymers. These materials exhibit intermediate properties between conventional liquids and solid crystals. The occurrence of spontaneous polarization is observable in liquid crystals that have a smectic phase, which consists of rod-shaped molecules (mesogens) with a center of chirality [74].

Researchers have developed ferroelectric liquid crystal polymers using bases such as polysiloxanes, polyethers, and polyesters, often incorporating side chains with more than one chiral center or several side-chain structures. These ferroelectric liquid crystal polymers exhibit a wide range of physical properties, making versatile materials with ferroelectric characteristics [75].

Over the past ten years, advanced fiber mats with exceptional functionalities and responsiveness have emerged by incorporating liquid crystals inside a polymeric matrix using electrospinning [74,76,77]. The liquid crystal component, responsible for the active features of the composite system, is commonly encapsulated in fibers with a core–shell structure, manufactured through coaxial electrospinning [78,79,80]. Alternatively, it can be an integral part of the polymer itself, as seen in liquid crystal elastomers [74,81,82].

### 3.2. Electrospinning-Assisted Ferroelectric Inorganic Nanofibers

Electrospinning has emerged as a versatile technique for producing nanofibers of inorganic ferroelectric materials. The process involves mixing appropriate reagents in precursor solutions along with a transport polymer [25,83]. The morphology and diameter of the nanofibers can be controlled by adjusting the concentrations of the reagents in the precursor solutions [22,25,51,83,84].

To obtain inorganic nanofibers, the prepared fiber mats must undergo a heat treatment process. This vaporizes the base polymer, promotes the coalescence and growth of the grains, and stabilizes the ferroelectric phase [51,55,85]. The heating rate must be carefully controlled to allow for slow polymer vaporization and gradual fiber formation. If the vaporization occurs too rapidly, the granular inclusions may not have sufficient time to connect, resulting in the formation of powders instead of fibers. Table 1 summarizes recent work on the production of inorganic ferroelectric nanofibers using this procedure, along with their respective heat treatments, purpose, and potential applications.

Among the many ferroelectric materials studied, barium titanate (BaTiO_3_, BTO) and lead zirconate titanate (PZT) [22,25,51,83,84,85,93,97,99,102,107,108] have received significant attention.

Barium titanate nanofiber mats have been produced with both aligned and randomly oriented nanofiber morphologies, as shown in Figure 2 and Figure 3. The fibers typically have average diameters ranging from 150 to 400 nm and consist of connected BaTiO_3_ nanoparticles forming long filaments.

Studies on these types of inorganic ferroelectric fibers have focused mainly on their synthesis and applications. Few studies have demonstrated the existence of ferroelectricity in fibers. Different phases can coexist at room temperature (cubic, tetragonal, orthorhombic), and the reduction to nanoscale tends to favor the stabilization of the paraelectric, non-ferroelectric cubic phase. From structural measurements and Raman spectroscopy measurements, the production of nanofibers with the tetragonal ferroelectric phase was observed for nanofibers annealed in the temperature range of 800–1000 °C [51,55,84,107,108]. Higher annealing temperatures led to better crystallization of the BaTiO_3_ nanofibers but also to rougher nanofiber surfaces, with optimized annealing temperatures occurring in the region of 900–1000 °C.

The temperature-dependent second harmonic generation response presented an anomaly in the vicinity of the paraelectric–ferroelectric phase transition (Figure 2d), confirming the change to the ferroelectric phase with decreasing temperature. The existence of domain structures and local switching, studied by piezoelectric force microscopy (PFM), presents clear evidence of the polar phase at room temperature. Thus, it was possible to determine the conditions for the preparation of ferroelectric fibers with high alignment and to demonstrate the presence of its ferroelectric phase at room temperature [85].

Taking advantage of the synthesis studies, more recently, barium titanate nanofiber mats have been used as components in the development of piezoelectric nanogenerators [90]. A piezoelectric nanogenerator with BaTiO_3_ electrospun nanofibers aligned vertically within polydimethylsiloxane (PDMS) was able to produce an output power of 0.1841 µW with maximum voltage of 2.67 V and current of 261.40 nA under a low mechanical stress of 2 kPa. Additionally, the system displayed a high dielectric constant of 40.23 at 100 Hz. The harvested energy was sufficient to power a commercial LED directly or, alternatively, could be stored in capacitors after rectification [90].

Barium titanate electrospun nanofibers have been used as piezoelectric sensors [102,109], as seen in Figure 4. These sensors incorporate barium titanate ceramic nanofiber mats, with a polymer-like softness of 50 mN, a large Young’s modulus of 61 MPa, and an elastic strain of 0.9%, resisting fracture after deformation. The nanofiber mats generated a highly sensitive piezoelectric response, producing an output voltage of 1.05 V when subjected to a pressure of 100 kPa, with a response time of 80 ms.

Beyond pure BaTiO_3_ fibers, doped barium strontium titanate (Ba_0.6_Sr_0.4_TiO_3_ or BST) [98,99] nanofibers have also been produced with diameters in the range of 100–200 nm. BST is attractive due its large dielectric constant, high tenability, low dc leakage, low loss tangent, and stable operation at high temperature. Higher calcination temperatures promoted improved BST nanofiber crystallinity [98]. The incorporation of BST fibers in a polymeric matrix has been explored for use as nanocapacitors. High energy densities and fast discharge speeds, with improved dielectric permittivity, were obtained, making them potentially attractive for energy-storage applications [99].

Bismuth ferrite (BiFeO_3_) nanofibers have been developed for a variety of applications. Their ferroelectric properties were characterized after different annealing temperatures. Their photovoltaic performance displayed enhanced voltage generation compared to films [104]. The inclusion of electrospun BiFeO_3_ fibers inside ferroelectric polymers such as PVDF has been leveraged to produce flexible piezoelectric energy harvesters [103]. Additionally, the use of BiFeO_3_ nanofibers in photocatalysis was explored in [105]. They were able to more efficiently degrade pollutants than in the bulk form.

Lanthanum-doped bismuth titanate (Bi_3.25_La_0.75_Ti_3_O_12_, BLT) ferroelectric nanofibers with diameters of between 100 nm and 300 nm were fabricated by electrospinning followed by annealing at 700 °C in air for 1 h [110]. Piezoeresponse force microscopy (PFM) measurements of the strain-versus-voltage hysteresis loops indicated ferroelectricity and an effective piezoelectric coefficient of *d*_33_ = 61 pm/V, about three times greater than that of a pure BLT film.

Lead zirconate titanate (PZT) ferroelectric nanofibers have been developed for flexible, wearable, and self-powered energy harvesters and nanogenerators, taking advantage of their significant piezoelectric coefficient [5,95,96,105]. Their synthesis conditions—in particular, their annealing treatments—were studied in Ref. [111]. Typical fiber diameters range from 50 to 200 nm. Electrospun PZT piezoelectric nanogenerator fiber mats can deliver 1.63 V, with an output power of 0.03 μW [5], while oriented multilayer PZT fibers stacked together have achieved an ultra-high output voltage of 209 V and a current density of 23.5 μA/cm^2^ [95,96].

In conclusion, electrospinning-assisted synthesis of ferroelectric inorganic nanofibers offers a promising route for developing advanced materials with tailored properties for various applications in energy harvesting, sensing, and catalysis.

### 3.3. Electrospinning-Assisted Multiferroic Nanofibers

Beyond single-phase ferroelectrics, magnetoelectric (ME) multiferroic materials, which present simultaneous magnetic and electric ordering, have attracted significant scientific and technological interest [112]. These materials hold promise for applications in information storage, sensors, actuators, and low-power energy-efficient electronics [113,114,115,116,117]. However, in single-phase multiferroics, the microscopic mechanisms underlying ME coupling have been challenging to unravel, hindering the development of strong ME responses at useful temperatures [112]. Nevertheless, single-phase BiFeO_3_ electrospun nanofibers have been reported [103,104,105,106], exhibiting a high magnetoelectric coupling coefficient (α_33_) comparable to that of BFO bulk and thin films [106]. However, research on single-phase multiferroic electrospun nanofibers remains limited, highlighting the need for the development and characterization of other materials.

To address these limitations, artificial multiferroic composites combining ferromagnetic (magnetostrictive) and ferroelectric (piezoelectric) materials have emerged as a potential alternative to single-phase multiferroics. These composites present robust magnetoelectric properties [118,119,120], with ME coupling coefficients of up to ~100 mVcm^−1^Oe^−1^ [120]. The strong ME coupling arises from the mechanical interaction between both phases, linking the electric, magnetic, and elastic fields. By carefully selecting the interface geometry and constituent materials, the ME coupling in composites can be engineered and optimized.

Various geometries have been explored for multiferroic magnetoelectric nanostructures, including self-assembled magnetostrictive nanopillars in a piezoelectric matrix, multilayers, and magnetostrictive nanograins in a piezoelectric matrix [112,121]. However, in thin-film structures, substrate clamping can significantly constrain strain-mediated interactions, limiting the achievable ME properties [122]. Interestingly, it has been predicted that substrate-free structures, such as nanofibers, can exhibit ME properties several orders of magnitude higher than their thin-film counterparts [122,123].

Hybrid ferromagnet–ferroelectric nanofibers represent an exciting platform for such applications and can be prepared with various phase morphologies. These include core–shell structures (Table 2), Janus or side-by-side arrangements (Table 3), and randomly mixed composite configurations (Table 3). In core–shell nanofibers, the ferromagnetic material forms the core, surrounded by a ferroelectric shell. In Janus nanofibers, the ferromagnetic and ferroelectric phases are arranged side by side. In composite nanofibers, ferromagnetic and ferroelectric precursors are mixed into a single solution, creating a random distribution of the phases.

Core–shell fibers allow for the creation of multiferroic systems with increased interfacial interaction between the magnetic and ferroelectric phases, potentially increasing their magnetoelectric response. An interesting example is that reported by Liu et al. [124], who observed shifts in the ferromagnetic resonance provoked by an applied electric field. On the other hand, Janus nanofibers, characterized by two distinct phases side by side, offer the advantage of exposing both phases while maintaining a substantial interfacial contact area. This can allow for the efficient transfer of mechanical stress transfer between magneto-strictive and piezoelectric components, as reported by Mathew et al. [125], who combined magneto-strictive cobalt ferrite with piezoelectric barium titanate to construct a compact sensitive magnetic field sensor. Multiferroic composite fibers made by directly mixing the ferroelectric and magnetic phases in the precursor solutions aim to offer similar advantages as core–shell and Janus fibers while maintaining simpler preparation conditions.

All these architectures can be fabricated using electrospinning. Core–shell nanofibers are produced using a dual syringe pump and a coaxial needle. The solution for the magnetic core is pumped through the inner part of the needle, while the ferroelectric shell solution is simultaneously pumped through the outer part. Janus fibers can be fabricated with two needles placed side by side and pumping the ferromagnetic and ferroelectric solutions together. For composite fibers, the single mixed solution is pumped through one needle, as in conventional electrospinning.

These substrate-free, tunable nanofiber geometries provide a versatile platform to enhance magnetoelectric coupling and broaden the range of functional applications for multiferroic materials.

#### 3.3.1. Core–Shell Multiferroic Fibers

Core–shell composite nanofibers, featuring a magnetic core surrounded by a ferroelectric shell, are the most common type of multiferroic nanofibers produced by electrospinning. Various material combinations have been successfully employed to fabricate these nanofibers, including CoFe_2_O_4_–Pb(Zr_0.52_Ti_0.48_)O_3_ [126], NiFe_2_O_4_–Pb(Zr_0.52_Ti_0.48_)O_3_ [124,127], NdFeO_3_–Pb(Zr_0.52_Ti_0.48_)O_3_ [128], CoFe_2_O_4_–BiFeO_3_ (see Figure 5) [129,130], NiFe_2_O_4_–BiFeO_3_ [130], NiFe_2_O_4_–BaTiO_3_ [131], and CoFe_2_O_4_–Ba(Zr_0.2_Ti_0.8_)O_3_–0.5(Ba_0.7_Ca_0.3_)TiO_3_ [132], as summarized in Table 2. Figure 5 presents examples of some of these core–shell fibers.

**Table 2 nanomaterials-15-00409-t002:** Multiferroic core–shell nanofibers prepared by electrospinning. The transporting polymers and high-temperature procedure to vaporize the polymer and coalesce the grains to form the fibers are presented, along with the magnetoelectric coefficient, when measured. PVP—polyvinylpyrrolidone, PMMA—polymethyl methacrylate, PVA—polyvinyl alcohol, MD—magnetodielectric measurements.

Polymer	Core–Shell Fibers	Heat Treatment	Magnetoelectric Coefficient	Ref.
PVP for CFO Mw = 1,300,000 PMMA for PZT Mw = 120,000	CoFe_2_O_4_–Pb(Zr_0.52_Ti_0.48_)O_3_ CFO–PZT	Nanofibers were collected on Pt/Ti/SiO_2_/Si substrates and dried at 120 °C for 4 h, followed by heating at 400 °C and then thermal annealing at 750 °C for 2 h in air	α_31_ = 29.5 V/cmOe	[126]
PVP for NFO and PZT Mw = 1,300,000	NiFe_2_O_4_–Pb(Zr_0.52_Ti_0.48_)O_3_ NFO–PZT	Fibers dried at 40 °C for 24 h and annealed in air at 650 °C for 1 h	Av = δHr /V Av = −24 Oe/V Converse effect, @ 5.4 GHz [85] MD = Δε′/ε′ MD = 8% (unassembled), −2% (assembled in magnetic field) @ 20–22 GHz, H = 0.8 T [127]	[124,127]
PVP for NDO Mw = 1,300,000 PVA for PZT Mw = 50,000	NdFeO_3_–Pb(Zr_0.52_Ti_0.48_)O_3_ NDO–PZT	Nanofibers were kept on a hot plate to dry at 100 for 10 h, followed by annealing at 850 C for 8 h.	-	[128]
PVP for CFO and BFO	CoFe_2_O_4_–BiFeO_3_ CFO–BFO	Nanofibers were collected to Pt/Ti/SiO_2_/Si substrates and dried at 120 °C for 4 h, followed by thermal annealing at 750 °C for 2 h in air.	220–250 V/cm Oe -	[129]
PVP Mw = 1,300,000 For NFO and BFO	NiFe_2_O_4_–BiFeO_3_ NFO–BFO	The samples were dried at 80 °C for 8 h, calcined at 350 °C for 2 h, and then calcined at 700 °C for 4 h in air.	-	[130]
PVP Mw = 1,300,000 For NFO and BTO	NiFe_2_O_4_–BaTiO_3_ NFO–BTO	Dried in an oven at 40 °C for 24 h, and then annealed for 1 h at 600–700 °C in air	0.4 mV/cm Oe @ 30 Hz	[131]
PVP for CFO and BCZT	CoFe_2_O_4_–Ba(Zr_0.2_Ti_0.8_)O_3_–0.5(Ba_0.7_Ca_0.3_)TiO_3_ (CFO–BCZT)	Dried at 120 °C for 90 min and calcined at temperatures of between 700 °C and 1000 °C for 1 h	-	[132]
PVP for BNFO and PZT	Ba_2_Ni_2_Fe_12_O_22_–Pb(Zr_0.52_Ti_0.48_)O_3_ BNFO–PZT	Dried on a hot plate at 100 °C for 6 h and then annealed at 1100 °C for 6 h	MD = Δε′/ε′ MD = –2.4% @ 100 Hz, H = 0.6 T	[133]
PVP for CFO and BCTSO	CoFe_2_O_4_–Ba_0.95_Ca_0.05_Ti_0.89_Sn_0.11_O_3_ CFO–BCTSO	Fibers dried at 80 °C under vacuum for 12 h and then annealed at 700 °C for 4 h in air	α = 0.346 V/cm Oe @ H = 1 T	[134]

#### 3.3.2. Composite and Janus Multiferroic Fibers

Composite nanofibers with random mixing between both phases and Janus nanofibers have been successfully prepared by electrospinning in systems such as Ni_0.8_Zn_0.2_Fe_2_O_4_–Ba_0.7_Sr_0.3_TiO_3_ [135], CoFe_2_O_4_–BaTiO_3_ [136], CoFe_2_O_4_–Pb(Zr_0.52_Ti_0.48_)O_3_ [137,138], NiFe_2_O_4_–Pb(Zr_0.52_Ti_0.48_)O_3_ (see Figure 6) [139], CoFe_2_O_4_–(Ba_0.95_Ca_0.05_)(Ti_0.89_Sn_0.11_)O_3_ [140], and CoFe_2_O_4_–BiFeO_3_ [141]. A summary of recent work is presented in Table 3.

**Table 3 nanomaterials-15-00409-t003:** Multiferroic composite and Janus nanofibers prepared by electrospinning. The transporting polymers and high-temperature procedure to vaporize the polymer and coalesce the grains to form the fibers are presented, along with the magnetoelectric coefficient, when measured. PVP—polyvinylpyrrolidone, MOKE—magneto-optic Kerr effect, SHG—second harmonic generation.

Polymer	Composite and Janus Fibers	Heat Treatments	Magnetoelectric Coefficient	Ref.
PVP Mw = 1,300,000	Ni_0.8_Zn_0.2_Fe_2_O_4_–Ba_0.7_Sr_0.3_TiO_3_ NZFO–BSTO composite fibers	Composite BSTO/NZFO fibers with molar ratios of 95/5, 90/10, 80/20, and 70/30 were annealed at 700 °C for 2 h.	MD = Δε′/ε′ MD ~18.2% @ 1 kHz, H = 6.3 kOe	[135]
PVP	CoFe_2_O_4_–BaTiO_3_ CFO–BTO composite fibers	Composite CFO/BTO fibers with a molar ratio of 50/50 were dried in an oven at 120 °C for 8 h, followed by annealing at 700 °C for 2 h.	α_31_ = 7.8 V/cmOe	[136]
PVP Mw = 1,300,000	CoFe_2_O_4_–Pb(Zr_0.52_Ti_0.48_)O_3_ CFO–PZT composite fibers	Composite CFO/PZT fibers with molar ratios of 0.75:1, 1:1, and 1.25:1 were dried at 120 °C for 4 h⁠, followed by heating at 400 °C and then annealing at 550 °C for 2 h in air.	-	[137,138]
PVP Mw = 1,300,000	NiFe_2_O_4_–Pb(Zr_0.52_Ti_0.48_)O_3_ NFO–PZT composite fibers	Composite NFO/PZT fibers with molar ratios of 0.75:1, 1:1, and 1.25:1 were dried at 120 °C for 4 h⁠, followed by heating at 400 °C and then annealing at 550 °C for 2 h in air.	-	[139]
PVP Mw = 1,300,000	CoFe_2_O_4_–(Ba_0.95_Ca_0.05_)(Ti_0.89_Sn_0.11_)O_3_ CFO–BCTSO composite fibers	Composite BCTSn/CFO fibers with a 1:1 molar ratio were dried at 80 °C under vacuum for 12 h before being annealed at 700 °C for 4 h in air.	-	[140]
PVP Mw = 1,300,000	CoFe_2_O_4_–BiFeO_3_ CFO–BFO composite fibers	Nanofibers with molar ratios of 1:0, 1:0.5, 1:1, 1:1.5, and 0:1 were dried at 60 °C for 12 h, followed by heating at 400 °C for 1.5 h and then annealing at 600 °C for 2 h in air.	-	[141]
PVP Mw = 1,300,000	CoFe_2_O_4_–BaTiO_3_ CFO–BTO Janus fibers	The fibers were calcined at 750 °C for 2 h.	Changes in magnetization at the BTO ferroelectric Curie temperature [23,142,143]; MOKE [144] Magnetic field-dependent polarization-resolved SHG [145]	[23,142,143,144,145,146]

Janus-type biphasic nanowires or nanofibers, characterized by two distinct phases in their hemi-cylindrical sections, have been successfully synthesized [23,142,143]. Unlike core–shell structures, where the internal component is inaccessible, Janus-type fibers offer the advantage of exposing both phases while maintaining a substantial interfacial contact area. This unique configuration presents significant potential for enhanced strain coupling in multiferroic nanocomposites. Janus composites of CoFe_2_O_4_–BaTiO_3_ are particularly interesting, and other examples are also listed in Table 3.

Research on the magnetoelectric (ME) response of electrospun fibers with core–shell, composite, or Janus-type multiferroic structures remains limited. Most studies have primarily focused on processing methods and basic structural characterization, with less emphasis on comprehensive functional property analysis. Despite this, these structures have demonstrated promising multiferroic and magnetoelectric behavior, with ME coefficients ranging from 0.3 to 29.5 Vcm^−1^Oe^−1^ (see Table 2 and Table 3). This emerging field warrants further investigation.

**Figure 6 nanomaterials-15-00409-f006:**
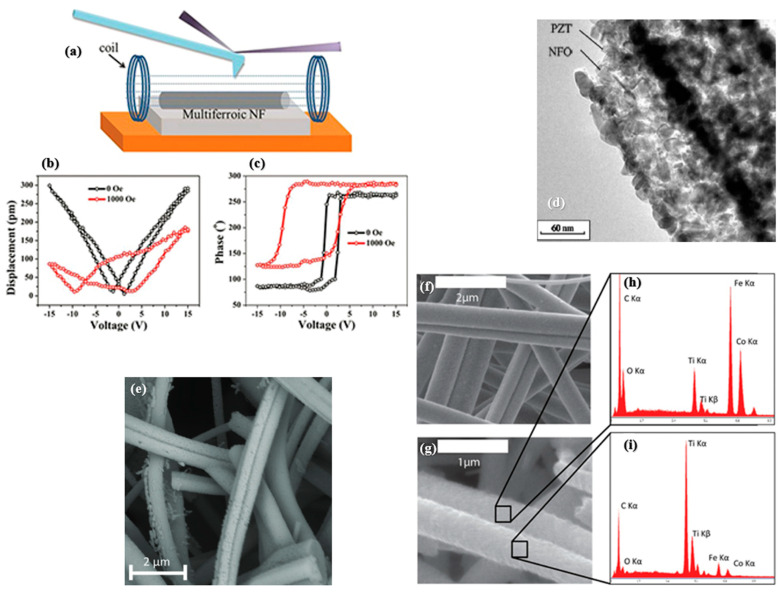
Magnetoelectric coupling of CFO–BTO composite nanofibers. (**a**) Conceptual diagram of the VFM investigation of multiferroic nanofibers with an external magnetic field. (**b**) Amplitude-voltage butterfly curves and (**c**) phase–voltage curves of the nanofibers with and without an external applied magnetic field. Reprinted with permission from Ref. [136]. (**d**) Composite NFO–PZT multiferroic nanofibers, reprinted with permission from Ref. [139]. (**e**) CFO–BTO Janus nanofibers, reprinted with permission from Ref. [145]. Characterization of calcined Janus nanofibers composed of CoFe_2_O_4_–BaTiO_3_. (**f**) A scanning electron microscope image (SEM) of a mat of randomly aligned Janus fibers. (**g**) An additional fiber from which further energy dispersive spectroscopy (EDS) spectra were obtained. (**h**) EDS spectra indicating the localization of CoFe_2_O_4_ to the darker semicylinder, and (**i**) EDS spectra confirming the localization of BaTiO_3_ to the lighter semicylinder. Reprinted with permission from Ref. [143].

## 4. Composite Fibers with Organic Ferroelectric Crystals

Beyond ferroelectric polymers, nanofibers of ferroelectric crystalline materials such as triglycine sulfate (TGS), (1,4-diazabicyclo[2.2.2]-octane), perrhenate (dabcoHReO_4_), and β-glycine have been developed in recent years [24,147,148]. TGS, a well-known ferroelectric material with a Curie temperature just above room temperature (TC~50 °C), is widely used in infrared detectors, taking advantage of its high pyroelectricity.

Nanofibers formed from TGS tend to assume a core–shell structure, where the core predominantly consists of the base polymer and the TGS grains form a shell surrounding it. Piezoelectric force microscopy (PFM) and impedance spectroscopy [147,149] established the presence of ferroelectricity in these fibers, with a ferroelectric transition temperature close to that of bulk TGS crystals.

Similarly, fibers were synthesized from dabcoHReO_4_ (1,4-diazabicyclo[2.2.2]-octane perrhenate), a crystalline ferroelectric material known for having the highest spontaneous polarization among water-soluble organic ferroelectrics. Structural, piezoelectric (PFM), and pyroelectric characterization [24] demonstrated their ferroelectric nature and revealed promising pyroelectric and piezoelectric coefficients, making them suitable for nano energy-harvesting applications. These fibers achieved a mechanical-to-electrical energy conversion efficiency of approximately 14% (Figure 7), underlining their potential for energy-harvesting applications.

Electrospun composite nanofibers functionalized with a benzothiazole derivative (BZT 1), incorporating an arylthiophene π-conjugated bridge, dispersed in a poly-L-lactic acid (PLLA) matrix, were successfully synthesized [150]. These fibers were characterized by their linear and nonlinear optical properties as well as their piezoelectric and ferroelectric behavior.

Evaluation of the first molecular hyperpolarizabity β revealed that BZT 1 is an efficient second harmonic generator. Local piezoelectric measurements confirmed that the heterocyclic system crystallizes in a nanocrystalline acentric structure when embedded in electrospun nanofibers. Measurements of piezoelectric hysteresis loops via PFM enabled the study of local domain switching under an applied external electric field. The observed out-of-plane hysteresis loop provided direct evidence for the switching of polarization in BZT 1 nanofibers and their ferroelectric properties.

The BZT 1 compound displays a high fluorescence quantum yield of 0.58 (Figure 8b). Figure 8d,e present the PFM images obtained from a single BZT nanofiber deposited on a conductive substrate. The first image shows the As-electrospun nanofiber before any external polarization field was applied. In the next two images (Figure 8d,e), the out-of-plane PFM contrast is observed after applying +100 V (Figure 8d) and then −100 V DC voltage (Figure 8e) to the conductive cantilever tip for 10 s. The pronounced contrast in the PFM images arises from an induced deformation in response to the applied ‘‘read’’ AC electric field and represents the components of the fiber’s polarization vector P. This contrast is roughly proportional to the longitudinal piezoelectric coefficient and is determined by the projection of the polarization vector normal to the substrate. The phase φ of the signal depends on the orientation of the polarization. ‘‘Bright’’ contrasts in the measured A. cos(φ) signal indicate that the polarization lies in the substrate plane, while the dark contrasts correspond to the polarization pointing downward to the surface.

The corresponding effective piezoelectric coefficient was estimated to be *d*_33_ = 20 pm/V, comparable to the piezoelectric coefficient of a poled poly(vinylidene fluoride) film and dabcoHReO_4_ nanofibers [24]. The switching of polarization confirms the ferroelectric nature of BZT 1, consistent with an acentric crystalline structure. Ferroelectricity is only possible in polar crystallographic point groups lacking a center of inversion [151].

Lead-free organic ferroelectric perovskite nanocrystals (MDABCO–NH_4_I_3_) were integrated into polymer fibers via electrospinning for mechanical energy harvesting [152], as illustrated in Figure 9. These molecular ferroelectrics, offering structural diversity and easy fabrication, exhibit piezoelectric and ferroelectric properties within the organic–inorganic hybrid materials category. The resulting flexible nanofibers act as effective piezoelectric energy harvesters, achieving an effective piezoelectric voltage coefficient g*_eff_* of 3.6 VmN^−1^ and emitting blue luminescence at 325 nm. During the ferroelectric–paraelectric phase transition, embedded MDABCO–NH_4_I_3_ perovskite nanocrystals display a pyroelectric coefficient comparable to state-of-the-art bulk ferroelectric materials. Moreover, these polymer-embedded nanocrystals maintain stable piezoelectric output, resisting oxidation-induced degradation. This durability, combined with their flexible nature, positions these nanocrystal systems as promising candidates for energy-harvesting applications.

## 5. Nonlinear Optical Nanofibers

Polymer-based nonlinear optical (NLO) and electro-optical (EO) materials offer an innovative alternative to conventional inorganic crystalline materials, such as lithium niobate (LiNbO_3_, LBO) and potassium dihydrogen phosphate (KH_2_PO_4_, KDP), for device applications. Polymeric materials possess several unique features that have driven research in the field. One notable advantage is their low spectral dispersion in the refractive index over a broad range of the electromagnetic spectrum. This characteristic enables the development of traveling-wave infrared modulators operating at exceptionally high modulation frequencies, exceeding 100 GHz, while maintaining relatively long interaction lengths (up to 1 cm). Another advantage of polymeric materials is that they facilitate the fabrication of robust, low-cost optoelectronic devices through monolithic integration with semiconductor electronics.

The electrical and optical properties of organic materials can be tailored through organic synthesis, allowing for the incorporation of chromophores with significant NLO properties. This results in materials with large EO coefficients and low drive voltages. Controlling molecular morphology and dipole orientation is critical for optimizing the performance of NLO organic devices. In polycrystalline materials with randomly oriented molecules, the net effective dipole moment is negligible, limiting their utility [153].

Electrospinning provides a technique to achieve strong molecular orientation within polymeric fibers, which significantly enhances their anisotropic properties. Organic molecules embedded in electrospun fibers can exhibit high degrees of molecular order, enabling their application in nonlinear optics [154].

An alternative method to force a macroscopic molecular orientation in polycrystalline NLO materials is the polling of polymers doped with NLO molecules by application of a strong external DC electric field. Here, the guest–host polymer is heated above its glass transition temperature and subsequently placed in a strong electric field before cooling. The molecular orientation of the NLO chromophores is then frozen during the subsequent cooling process. While effective, this method often results in significant chromophore relaxation after electric field removal, leading to decreased anisotropic properties.

In contrast, the electrospinning technique is a powerful tool for controlling the morphology and orientation of polymer chains and the embedded crystalline NLO molecules, mitigating relaxation issues. It can be employed with a variety of different polymers and molecular complexes [155].

In one of the earliest examples, Isakov et al. [21] obtained strongly anisotropic optical second harmonic generation based on highly aligned nanofibers of 2-methyl-4-nitroaniline C_7_H_8_N_2_O_2_, MNA) embedded in poly-L-lactic acid (PLLA) nanofibers fabricated by the electrospinning technique (Figure 10a). MNA, a derivative of 4-nitroaniline or para-nitroaniline (C_7_H_8_N_2_O_2_, pNA), achieves noncentrosymmetric crystallization by substituting a hydrogen atom with a methyl group at the 2-position of the benzene ring. Crystals of MNA exhibit a very large phase-matched second harmonic generation (SHG) with a figure of merit over two orders of magnitude greater than that of lithium niobate [153].

X-ray diffraction analyses revealed that MNA molecules within the electrospun fibers predominantly crystallized with their (010) crystallographic planes parallel to the fiber plane (Figure 10e). Second harmonic generation polarimetry, shown in Figure 10f,g, confirmed a high degree of anisotropy in an array of MNA_PLLA electrospun nanofibers, with an effective nonlinear susceptibility coefficient estimated to be 148 pm/V. The signal generated by a mat of MNA_PLLA fibers is as strong as in bulk MNA, highlighting the potential of these nanofibers for a variety of NLO applications. Furthermore, the NLO response was strongly correlated with the size of the MNA nanocrystals within the polymeric fiber host, which could be tuned by adjusting the electrospinning processing parameters. Over the range explored, the nonlinear optical response of the electrospun nanofibers of MNA varied by one order of magnitude [156].

High-resolution confocal laser scanning microscopy (Figure 11a,b) demonstrated that MNA nanocrystals are homogeneously distributed within the PLLA polymer fiber, with particle sizes below 100 nm [156]. The mean MNA crystallite size varies with the fiber diameter: thinner fibers contain smaller crystallites. The SHG efficiency of the nanofibers varied with fiber diameter, showing reduced efficiency when the diameter approached the optical coherence length due to phase mismatch effects (Figure 11c).

Similar advances have been made using para-nitroaniline (pNA), a prototype molecule for organic NLO materials. It has a delocalized π–π* electron system and unsaturated bridge linking a donor amino group (NH_2_) and an acceptor nitro group (NO_2_). Despite its large molecular hyperpolarizability [157,158,159], bulk pNA crystals belong to the centrosymmetric space group P2_1_/n, preventing macroscopic second harmonic generation [160]. Embedding pNA into electrospun fibers overcame this limitation, as the process induced highly oriented mesocrystalline structures with a dominant acentric surface, as shown in Figure 12 [161,162].

During the electrospinning process, *p*NA was incorporated into PLLA polymer fibers with a strong preferred crystallographic orientation. The X-ray diffraction pattern was dominated by the (202) Bragg reflection, which was two orders of magnitude more intense than all the other observed reflections. However, all Bragg reflections could be assigned to the known centrosymmetric crystalline structure, indicating that it was not a polymorph.

NLO measurements performed on a single fiber concluded that the effective mean nonlinear optical coefficient d_eff_ was around 42 pm/V, twice that of the highest coefficient measured on meta-nitroaniline or 3-nitroaniline (mNA or 3NA) crystal, *d*_33_ = 21 pm/V, an established SHG crystalline compound. The results demonstrate the possibility of engineering a nanofiber array of centrosymmetric nanocrystals with a strong SHG response comparable to that of single non-centrosymmetric organic crystals. The observed effect resulted from the multiple crystal surfaces having a common orientation, with highly aligned molecules forming a head-to-tail dipolar arrangement. Fluorescence lifetime microscopy (Figure 12c) indicated the presence of strong excitonic coupling between molecular units within the fibers and also in bulk pNA crystals.

Another example involves 3-nitroaniline (3NA) nanocrystals embedded in poly-ε-caprolactone (PCL) 3NA@PCL fibers, displayed in Figure 13. The X-ray diffractogram of a 3NA@PCL nanofiber mat revealed that the (400) Bragg peak was the most intense, a notable contrast to the corresponding diffraction pattern from a bulk polycrystalline sample (Figure 13a). This indicates a strong preferential orientation within the fiber mat (Figure 13b). In the crystal unit cell, 3NA molecules are organized such that their molecular dipole moments (represented by arrows) collectively align to produce a net dipole moment parallel to the (400) plane, orientated along the polar axis. In the fibers, the (400) plane aligns parallel to the surface of the fiber mat surface so that the polar axis is directed along the longitudinal fiber axis. This preferential orientation arises due to the strong electric field applied during the electrospinning process, which aligns the high-molecular-dipole moments within the fibers.

The anisotropy of polarity curves measured on a nanofiber array and on a (100)-oriented 3NA single-crystal platelet, under identical excitation conditions, confirms the preferred orientation of the nanocrystals embedded into the fibers (Figure 13c). Notably the effective second-order susceptibility measured on a single nanofiber was $d_{eff}^{3NA@PCI}=80\;pm/V$, four times greater than $d_{eff}^{3NA}=21\;pm/V$, the corresponding value for macroscopic 3NA crystals [26]. Consequently, the push–pull nitroaniline derivatives embedded within each fiber act as enhanced, polarized nanoemitters of second harmonic generation (SHG) light. Furthermore, the nanocrystals within the fibers are significantly smaller than the optical coherence length of a 3NA platelet crystal (approximately 10 μm), enhancing the SHG output intensity compared to bulk 3NA since phase mismatch is insignificant in the fibers.

Beyond their nonlinear optical (NLO) properties, 3NA@PCL nanofiber mats also function as electric power sources via the piezoelectric effect. Applying a periodic force induces polarization within the acentric crystalline material, enabling an external electric current to flow between the opposite faces of the mat through a load resistance (Figure 13g,h). As illustrated in Figure 13g, a periodic force of 3 N generated a voltage and current of up to 7 V and 70 nA, respectively, yielding an instantaneous power density of 122 nW/cm^2^ [26].

Another push–pull molecule closely related to *p*NA is 2-amino-4-nitroaniline, (2A4NA, C_6_H_7_N_3_O_2_), which, like MNA, is an engineered molecule obtained by replacing a meta-hydrogen on the benzene ring by an amino group [163]. The NLO response of electrospun fiber mats composed of 2A4NA grains inside PLLA polymeric fibers (see Figure 14) were investigated and compared with 3NA crystals under identical conditions [164].

The average diameter of the electrospun 2A4NA@PLLA fibers was 770 nm. X-ray diffraction (XRD) patterns revealed a preferred orientation of 2A4NA and MNA nanocrystals within the polymer fibers. This orientational order was further confirmed by the anisotropy observed in SHG polarimetry curves. For both 2A4NA@PLLA and 3NA@PLLA, the q–p configuration yielded a maximum intensity nearly an order of magnitude higher than the q–s configuration. For comparison, Figure 13e shows the polarimetry curves measured on an individual 3NA@PCL fiber under the same excitation conditions [26]. Both the 3NA@PLLA and the 3NA@PCL fibers exhibited nearly identical anisotropic SHG responses, suggesting that the polymer matrix does not significantly alter the orientational distribution of the nanocrystals. There is negligible optical damage of the nanofibers during the acquisition, as the curves possess symmetric maxima and close on themselves. The nanofibers are optically robust to laser damage.

The estimated effective second-order susceptibility coefficient for 3NA@PLLA was $d_{eff}^{3NA@PLLA}=56\;pm/V$, which is 1.5 times smaller than that measured for 3NA@PCL nanofibers, $d_{eff}^{3NA@PCL}=80\;pm/V$ [26]. For 2A4NA@PLLA, the estimated effective second-order susceptibility coefficient was $d_{eff}^{2A4NA@PLLA}=42\;pm/V$, which is among the highest values reported for organic crystals formed by simple pNA-derivative push–pull molecules. The spatial distribution of second harmonic emitters within single 3NA@PLLA and 2A4NA@PLLA fibers is displayed in Figure 14c,d. The 3NA@PLLA fibers show a relatively uniform distribution of SHG emission along the excited fiber, whereas the SHG emission from a 2A4NA@PLLA fiber is punctuated by intense spots.

Finally, in addition to charge transfer push–pull molecules, electrospun nanofibers embedded with urea, a well-known organic crystalline material, have been produced from precursor solutions with two different polymers: poly(ethylene oxide) (PEO) and poly(vinyl alcohol) (PVA). Both polymers are very soluble in water, as is urea.

Urea was one of the first organic NLO crystals available commercially. Its nonlinear optical properties have been extensively studied due to its simple structure and strong second harmonic generation (SHG) response. In its bulk crystalline form, urea exhibits an effective second-order nonlinear coefficient approximately 2.5 times greater than that of potassium dihydrogen phosphate (KDP), a widely used inorganic crystal.

With PEO, highly crystalline and orientated nanofibers containing two different polymeric host–guest inclusions were formed (Figure 15). They are the stable α complex and the metastable β complex [165,166].

The α complex has a (PEO)4-(urea)9 stoichiometry [167], and fibers were successfully prepared from a gel-like suspension of a previous inclusion compound (IC) co-crystallized complex. These highly oriented and well-aligned fibers displayed a strong molecular orientation, with PEO chains confined inside urea channels. This strong molecular orientation inside the fibers was attributed to a reorientation of rigid crystalline units during the electrospinning process [168]. Remarkably, the electrospinning process stabilized the metastable β phase with a (PEO)3-(urea)2 stoichiometry, resulting in an orthorhombic crystal structure, as determined from the X-ray diffraction data [166].

**Figure 15 nanomaterials-15-00409-f015:**
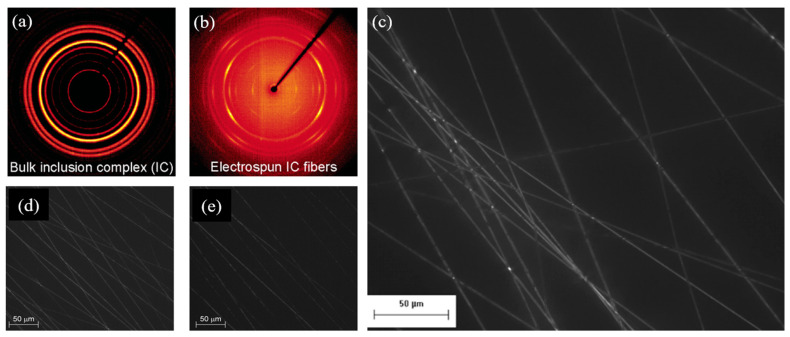
(**a**) Wide-angle X-ray diffraction 2θ diagrams for (**a**) the bulk PEO−urea inclusion complex (IC) and (**b**) electrospun fibers. (**c**) Cross-polarized optical micrograph of electrospun fibers of the PEO−urea inclusion complex. Reprinted with permission from Ref. [168]. Representative cross-polarized optical micrographs of fibers prepared by electrospinning solutions with (**d**) 4:9 and (**e**) 3:2 PEO:urea molar ratios. Reprinted with permission from Ref. [166].

Unlike the PEO–urea fibers, the PVA–urea nanofibers (1:1 polymer-to-urea mass ratio) showed no evidence of inclusion complexes in their X-ray diffraction patterns. Instead, urea nanocrystals inside the PVA matrix retained the tetragonal structure of bulk urea crystals (Figure 16). However, the nanocrystalline urea within the fibers displayed a strong preferential orientation with their (110) plane parallel to the fiber mat plane (Figure 16c).

The SHG response of urea was quite different in these two polymer fibers. In urea–PVA nanofibers, the urea nanocrystals generated a SHG response as intense as a pure urea powder, with an average grain size of 110 µm. The SHG response of isotropic PEO–urea fibers was about two orders of magnitude less than the urea–PVA fibers (Figure 16d,e). However, no SHG signal was observed from the urea–PEO fibers. These findings highlight the importance of the host polymer and the resulting crystal structure in determining the nonlinear optical properties of urea-embedded nanofibers.

Although the above results are encouraging, further research is needed to elucidate the mechanisms behind the differing SHG responses and to optimize the fabrication processes for enhanced nonlinear optical properties of organic nanocrystals embedded in polymeric electrospun fibers. Deeper insights into the process will facilitate the development of novel materials with tailored nonlinear optical characteristics.

## 6. Perspectives for Applications: Stability, Biocompatibility, Scalability, and Environmental Impact

Recent advances in the development of electrospun ferroelectric and nonlinear optical nanofibers have demonstrated their potential for a wide range of applications, including electronics, photonics, sensors, and biomedical devices. It has been forecasted that the global market for polymeric nanofibers should grow from USD 2.2 billion in 2021 to USD 6.7 billion by 2026 with a compound annual growth rate of 25.1% for the period of 2021–2026 [169,170]. However, for these materials to continue to improve their transition from laboratory research to commercial applications, several critical aspects must be considered, including their long-term stability, biocompatibility, large-scale production feasibility, and environmental impact.

### 6.1. Long-Term Stability

The long-term performance of electrospun nanofibers is an important factor in determining their viability for industrial and biomedical applications. Their stability can be affected by humidity, temperature fluctuations, and mechanical stress. In the case of inorganic ferroelectric nanofibers, degradation mechanisms such as oxygen content can compromise functionality over time. As inorganic nanofibers tend to be brittle, they have been successfully included in polymers, which gives them flexibility, strength, and long-term stability [99,103,105,109]. On the other hand, organic-based ferroelectric and non-linear optical nanofibers exhibit stable molecular organization, as they do not undergo rearrangements that could diminish performance. Organic inclusions, when embedded into electrospun nanofibers, maintain their properties for more than one year, as the polymer matrix acts as a shield by inhibiting oxidation [152,171,172]. Additionally, when non-centrosymmetric organic molecules crystallize within nano- and microfibers, they maintain their structural integrity, mitigating the risk of molecular reorganization. The electrospinning process effectively preserves molecular orientation, ensuring that organic molecules form single-crystal-like structures that keep their optical and piezoelectric properties and enhance their non-linear optical properties [21], which are important for photonic applications.

### 6.2. Biocompatibility

Biocompatibility is a key consideration for applications in biomedical engineering, including biosensors, tissue scaffolds, and implantable devices. Certain ferroelectric polymers, such as polyvinylidene fluoride (PVDF) and piezoelectric polymers like poly-L-lactic acid (PLLA), have demonstrated biocompatibility [173,174], making them suitable for use in biomedical applications. By embedding into those polymers non-toxic all organic molecules such as urea and β-glycine, as well semi-organic lead-free perovskites retaining their nonlinear optical, piezoelectric, and pyroelectric properties, it is possible to manufacture disposable, environmentally safe active electrospun fiber mats to be integrated into appropriate devices. On the other hand, inorganic ferroelectric nanofibers may require additional surface modifications to minimize cytotoxicity. Additionally, lead-free ferroelectric compounds are favored due to their reduced toxicity. In this respect, important ferroelectrics, such as lead-free BaTiO_3_, have shown very good biocompatibility, even having been used to coat other, more toxic compounds, in the context of their use in biomedical applications [175]. Ferrites such as CoFe_2_O_4_ have also shown good biocompatibility, particularly when embedded inside polymeric fibers [176,177]. Further in vitro and in vivo studies are necessary to evaluate the long-term interactions of these nanofibers with biological tissues.

### 6.3. Scalability of Production

For practical implementation, the scalable production of electrospun nanofibers with consistent properties is essential. Electrospinning offers significant advantages in terms of tunability and ease of processing. However, challenges remain in achieving uniform fiber morphology and composition when moving from laboratory-scale to industrial-scale production [178]. High-throughput electrospinning techniques, such as multi-jet, could provide solutions for scaling up manufacturing while maintaining the functional properties of the nanofibers [178]. Notable examples of multiple-needle electrospinning includes the Nanospinner developed by Inovenso Inc., which involves 110 needles and is used in the commercial production of filtration membranes and medical devices [179]. Needleless electrospinning has also found commercial use for high-throughput nanofiber production, such as the Nanospider Production Line developed by Elmarco Inc. (Liberec, Czech Republic) [180], which uses a wire electrode to eject multiple jets. Nevertheless, the reproducibility of fiber alignment, deposition, and integration into devices must continue to be optimized for commercialization.

### 6.4. Environmental Impact

As sustainability becomes a growing concern, the environmental impact of nanofiber production and disposal must be assessed [169]. Some ferroelectric and nonlinear optical nanofibers contain lead-based materials or organic molecules (some examples are included in Table 1, Table 2 and Table 3), which raise concerns regarding toxicity and recyclability [181]. Developing lead-free alternatives, such as lead-free barium titanate-based ferroelectrics [182,183], semi-organic perovskites, and organic molecules based on amino acids and peptides, can provide more environmentally friendly solutions [181,184]. Some of these solutions have been demonstrated in the present review. The incorporation of functionalized nano- and microfiber mats into appropriate devices contributes to material resource sustainability. This is because the embedded functional materials are in the form of powders, which are dissolved into a polymer matrix. Therefore, there is no need for using bulk materials, which are expensive to produce and involve extra resources. Additionally, the selection of biodegradable polymers and green solvents for electrospinning can reduce the ecological footprint of nanofiber production. Future research should focus on life cycle assessments and end-of-life strategies to ensure that these materials align with sustainable development goals.

## 7. Conclusions

In conclusion, this review highlights the promising potential of the electrospinning technique in fabricating multifunctional ferroelectric-based nanofibers from a diverse array of materials, including inorganic, semi-organic, and all-organic compounds. By carefully selecting an appropriate polymer matrix, these materials can be transformed into nano- and microstructure hybrid functional systems. These hybrid systems encompass a range of functional materials, including piezoelectric ceramics, inorganic ferroelectrics, single-phase ferroelectrics, and magnetoelectric multiferroic materials, which exhibit simultaneous magnetic and electric ordering. Among semi-organic crystals, particular emphasis was placed on triglycine sulfate (TGS) due to its technological significance.

In the realm of nonlinear optics, the fabrication of nanomaterials in the form of nanofibers using chromophores with high molecular hyperpolarizabilities offers an innovative alternative to conventional inorganic crystalline materials for device applications. In particular, nanofibers incorporating push–pull nitroaniline derivatives have demonstrated enhanced optical second harmonic generation and have strong potential as nanoscale emitters of polarized light.

Additionally, crystalline hybrid nanofiber mats with pronounced piezoelectric properties have been developed that are capable of efficiently converting modest applied forces into significant electrical outputs, generating appreciable piezoelectric voltages.

Looking forward, the quest for cost-effective sensors, actuators, and electronic and photonic components, alongside the increasing demand for device miniaturization and integration, positions electrospun nanofibers as a promising solution. The integration of multiple functionalities into a single nanostructure using the electrospinning technique allows researchers to address the key requirements of affordability, flexibility, and versatility when seeking to develop novel multifunctional device technologies.

## Figures and Tables

**Figure 1 nanomaterials-15-00409-f001:**
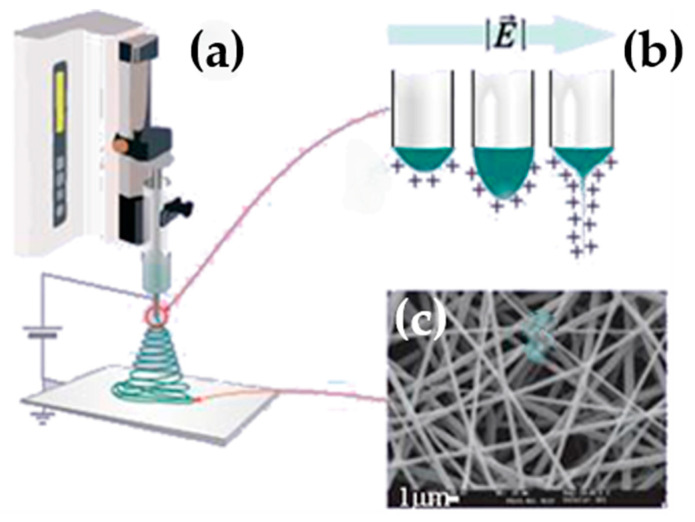
Electrospinning process, with (**a**) the typical experimental setup, (**b**) the Taylor cone formation and (**c**) the resulting fibers.

**Figure 2 nanomaterials-15-00409-f002:**
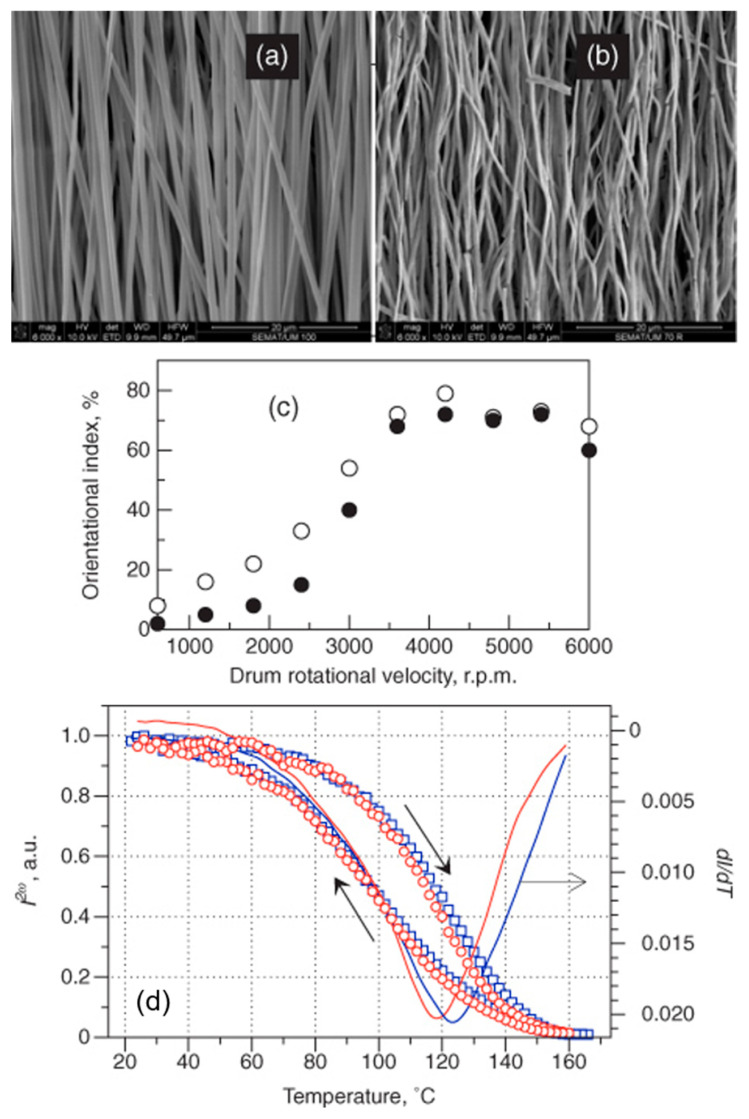
SEM images of the barium titanate fibers (**a**) As-electrospun and (**b**) after annealing. Scale bar: 20 µm. (**c**) Evolution of the alignment degree (orientational index) of the nanofibers, with the rotation velocity of the collector drum. Open and filled circles correspond to As-electrospun and annealed nanofibers, respectively. (**d**) The normalized SHG intensity as a function of temperature observed in nanofibers of BaTiO_3_ calcined at 900 °C (squares) and 1100 °C (circles). The solid curves are the derivatives of I(T), showing a minimum at the ferroelectric transition temperature of bulk BaTiO_3_. Reprinted with permission from Ref. [85].

**Figure 3 nanomaterials-15-00409-f003:**
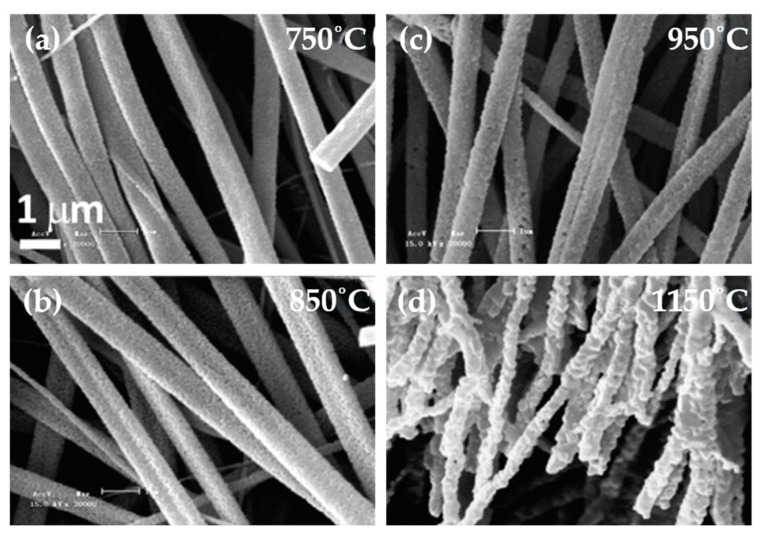
SEM images of the BaTiO_3_ nanofibers calcined at (**a**) 750 °C, (**b**) 850 °C, (**c**) 950 °C, and (**d**) 1150 °C for 6 h. All images have the same scale. Reprinted with permission from Ref. [108].

**Figure 4 nanomaterials-15-00409-f004:**
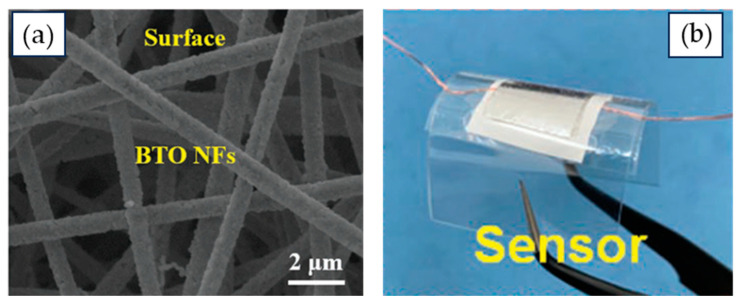
(**a**,**b**) Barium titanate flexible nanofiber mat. Reprinted with permission from Ref. [109].

**Figure 5 nanomaterials-15-00409-f005:**
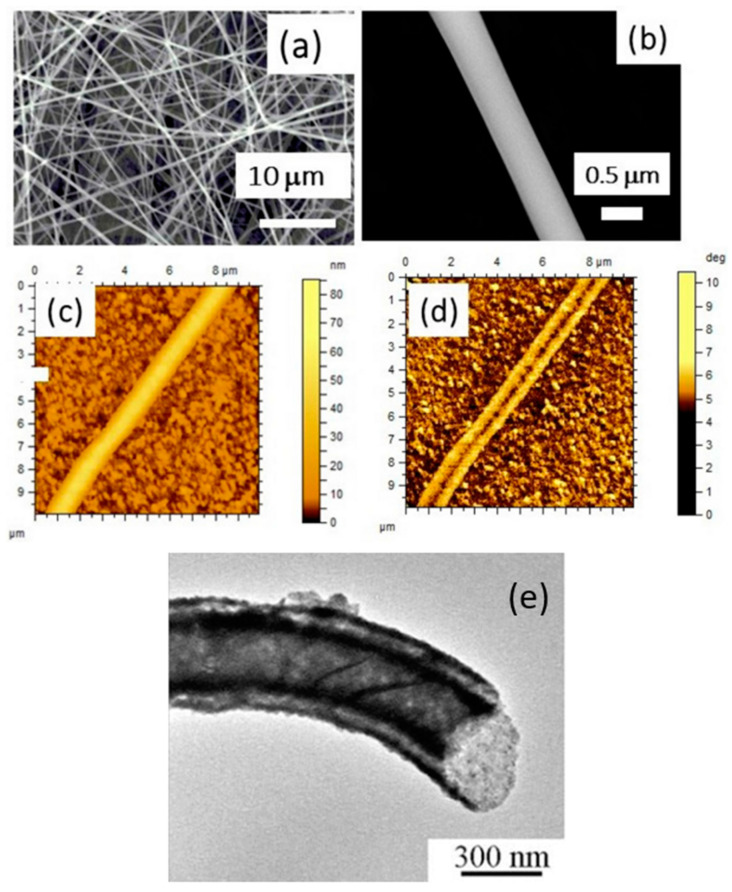
Scanning electron microscopy (SEM) micrograph of (**a**) coaxial fibers of sample A with a barium titanate (BTO) core and a nickel ferrite (NFO) shell and (**b**) a single fiber of sample A. (**c**) Atomic force microscopy (AFM) topography for fibers of sample B with an NFO core–BTO shell. (**d**) Magnetic force microscopy (MFM) phase image for sample B. Reprinted with permission from Ref. [131]. (**e**) Core–shell nanofibers composed by a CFO core inside a BFO shell. Reprinted with permission from Ref. [129].

**Figure 7 nanomaterials-15-00409-f007:**
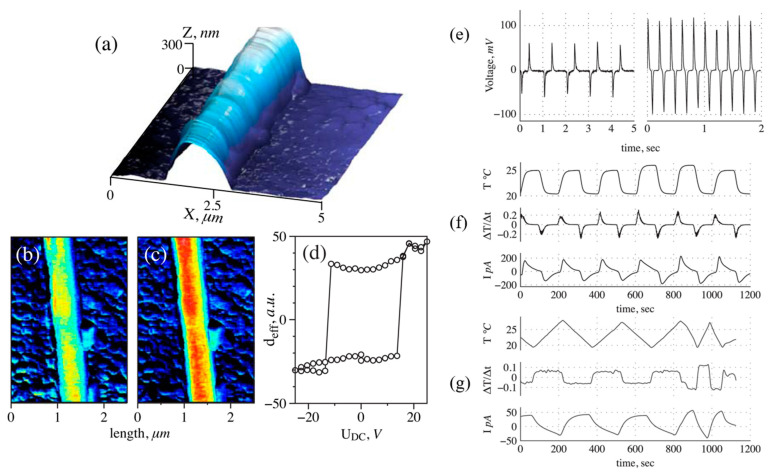
(**a**) Morphology of the dabcoHReO_4_ individual fiber obtained by AFM; (**b**) vertical and (**c**) lateral PFM images of dabcoHReO_4_ nanofibers; (**d**) piezoelectric response hysteresis loop; (**e**) voltage output of the fabricated nanofiber mat under 1 Hz (left) and 5 Hz (right) repeated compressive impacts. Nanofiber mat thickness of 100 μm, working area of 1 cm^2^. The temperature cyclic change, and derivative and output pyroelectric current obtained in dabcoHReO4 nanofiber mat at pulsed (**f**) and ramp (**g**) temperature changes. Reprinted with permission from Ref. [24].

**Figure 8 nanomaterials-15-00409-f008:**
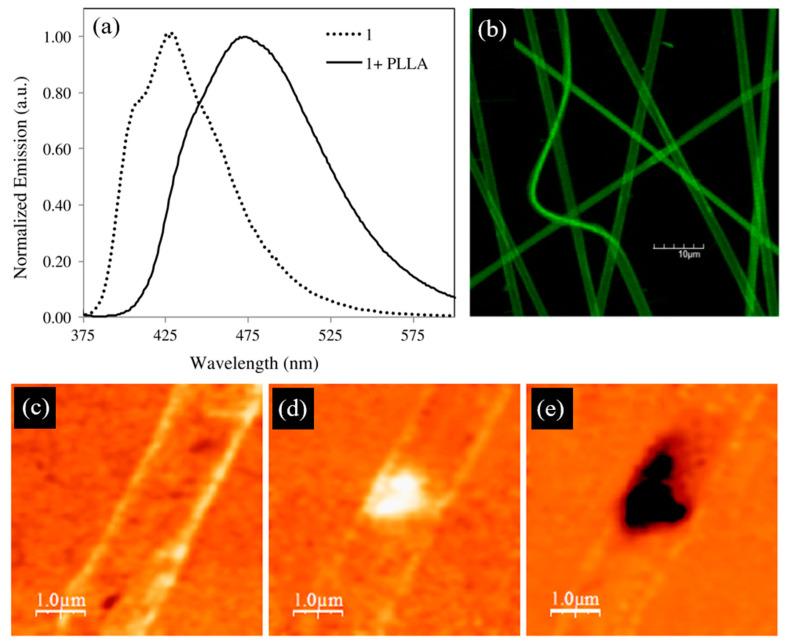
(**a**) Normalized emission spectra of BZT nanofibers and the pure compound in absolute ethanol; (**b**) single-photon fluorescence image (50 × 50 µm) of BZT 1 nanofibers. The fluorescence was observed in the wavelength range of 430–470 nm following excitation at a wavelength of 405 nm. Vertical PFM images observed in an individual BZT nanofiber: (**c**) As-electrospun nanofibers (before poling), (**d**) PFM image after applying a voltage of +100 V at the fixed tip during 10 s, (**e**) image after applying a voltage of −100 V. Reprinted with permission from Ref. [150].

**Figure 9 nanomaterials-15-00409-f009:**
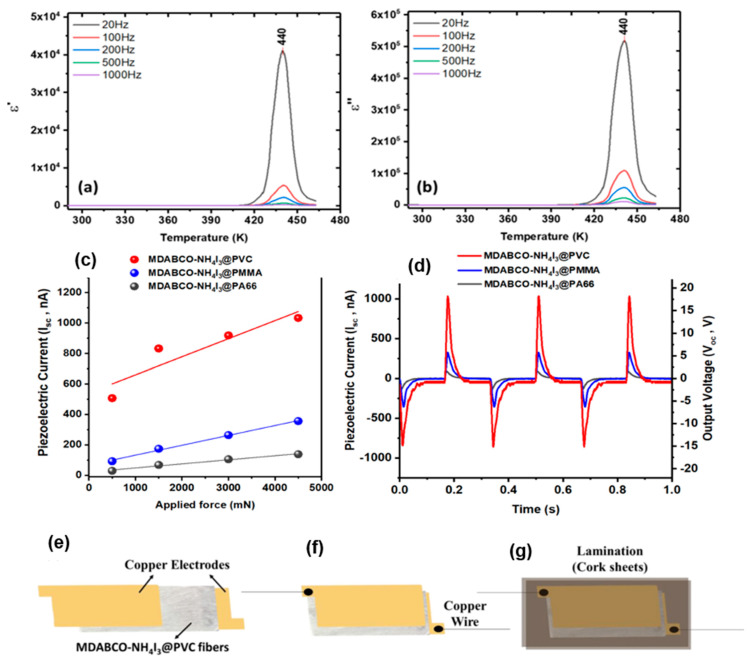
(**a**) Dielectric permittivity of an MDABCO–NH_4_I_3_ polycrystalline sample showing its (**a**) real and (**b**) imaginary parts as functions of temperature and frequency. The ferroelectric–paraelectric phase transition occurs at 440 K; (**c**) piezoelectric current as a function of the applied forces and (**d**) output voltage and current as a function of time from MDABCO–NH_4_I_3_ incorporated into different electrospun polymer nanofibers; (**e**–**g**) schematic of MDABCO–NH_4_I_3_@PVC piezoelectric nanogenerator, reprinted with permission from Ref. [152].

**Figure 10 nanomaterials-15-00409-f010:**
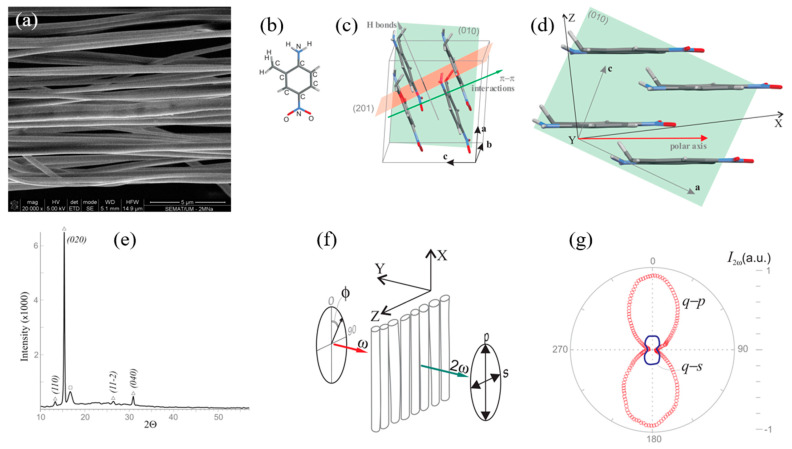
(**a**) SEM image of MNA@PLLA nanofiber arrays; (**b**) the MNA molecule; (**c**) MNA molecular packing and (**d**) its projection onto the (010) plane (**d**); (**e**) XRD pattern of MNA_PLLA fiber arrays; (**f**,**g**) sketch of the SGH ellipsometry experiment and SHG polarization patterns in the in-plane aligned array of the MNA@PLLA fibers, reprinted with permission from Ref. [21]. Here, the generated second harmonic light, which passed through an analyzer, was oriented either parallel (*q*–*p* configuration, *q* indicates an incident polarization angle) or perpendicular (*q*–*s* configuration) to the longitudinal axis of the aligned fibers.

**Figure 11 nanomaterials-15-00409-f011:**
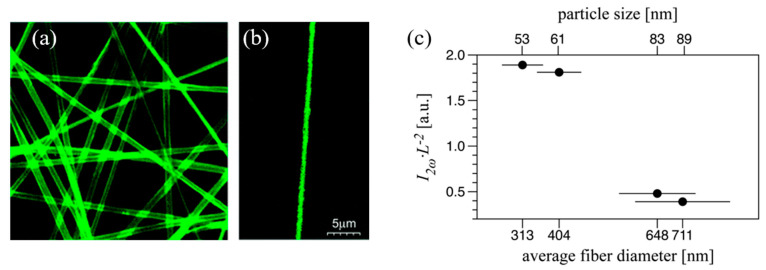
(**a**,**b**) Single-photon fluorescence image of MNA–PLLA nanofibers showing a uniform distribution of MNA nanocrystals embedded within the PLA nanofiber. The excitation wavelength was 405 nm, and fluorescence was observed in the range of 430–470 nm. (**c**) SHG efficiency of MNA–PLLA nanofibers as a function of the average fiber diameter and MNA particle size. Reprinted with permission from Ref. [156].

**Figure 12 nanomaterials-15-00409-f012:**
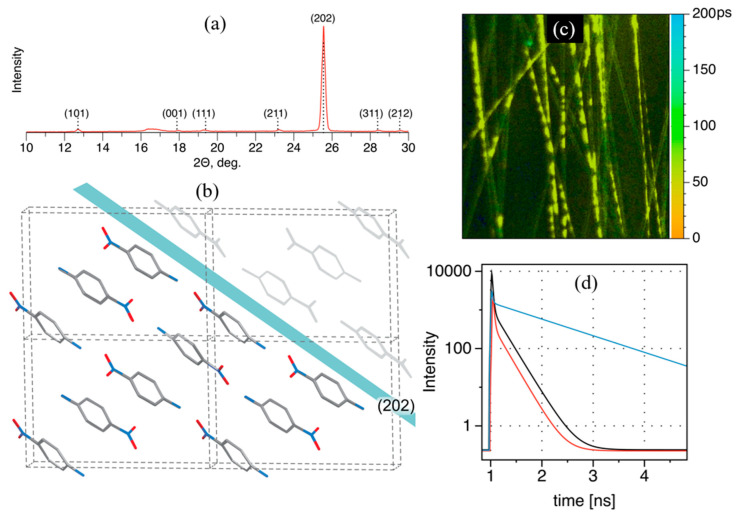
(**a**) X-ray diffraction pattern revealing strong orientation observed in a *p*NA mesocrystalline structure produced by electrospinning; (**b**) four-unit cells of *p*NA with the shown (202) plane terminated the periodic crystal lattice by the surface; (**c**) 50 × 50 μm micrograph of *p*NA nanofibers obtained by fluorescence lifetime imaging microscope; (**d**) double exponential fluorescence decay obtained from *p*NA single crystal (black curve), nanofibers (red curve), and solution (blue curve). Reprinted with permission from Ref. [161].

**Figure 13 nanomaterials-15-00409-f013:**
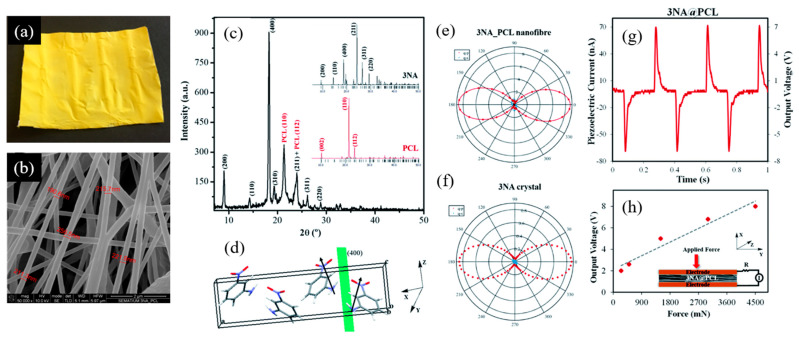
(**a**) 3NA@PCL electrospun fiber mat deposited on a substrate; (**b**) corresponding SEM image; (**c**) measured X-ray diffraction pattern of a 3NA@PCL nanofiber mat (the insets show the calculated powder patterns for crystalline 3NA and PCL polymer); (**d**) unit cell of 3NA showing the molecular dipoles (represented by arrows) added to a net dipole parallel to (400); (**e**) polar plot of SHG polarimetry data collected on a single 3NA@PCL electrospun nanofiber for *q–p* and *q–s* configurations; (**f**) polar plot of SHG polarimetry data collected on a (100) 3NA crystal platelet for *q–p* and *q–s* configurations. The radial axis values are expressed in the unit of counts. The maximum intensity corresponds to the case where the polarization of incident and emitted light are parallel to each other and aligned with the fiber longitudinal axis; (**g**) output voltage and current measured on a 3NA@PCL electrospun fiber mat; (**h**) plot of output voltage versus applied force with a schematic piezoelectric setup of a 3NA@PCL fiber mat. There is a linear relationship between the measured output voltage and the applied force. Reprinted with permission from Ref. [26].

**Figure 14 nanomaterials-15-00409-f014:**
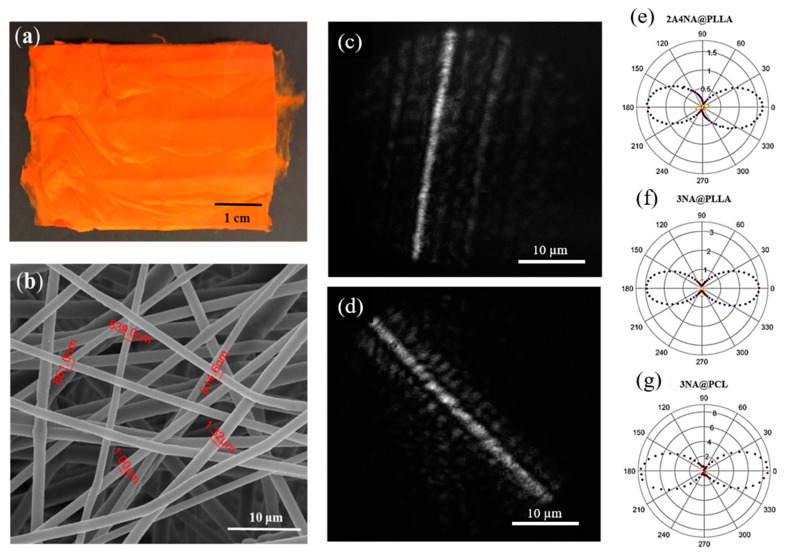
(**a**) Electrospun 2A4NA@PLLA fiber array deposited on a substrate, (**b**) corresponding scanning electron microscope image. Monochromatic images of the second harmonic light emitted by single fibers of (**c**) 3NA@PLLA and (**d**) 2A4NA@PLLA. Polar plots of SHG polarimetry data collected on a single nanofiber of (**e**) 2A4NA@PLLA, (**f**) 3NA@PLLA and (**g**) 3NA@PCL [26] for *q*–*p* and *q*–*s* configurations. The radial axis values are expressed in units of 106 counts. Reprinted with permission from Ref. [164].

**Figure 16 nanomaterials-15-00409-f016:**
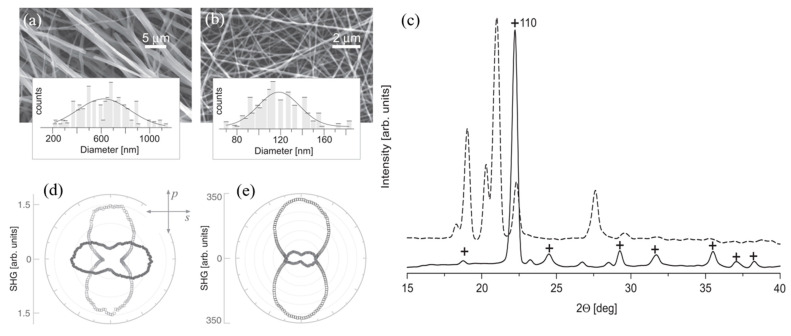
(**a**) SEM micrographs and thickness distribution of electrospun PEO–urea and (**b**) PVA–urea nanofibers. (**c**) X-ray diffraction patterns of electrospun fibers of PEO–urea (dashed line) and PVA–urea (solid line). Crosses indicate diffraction peaks of pure crystalline urea. (**d**,**e**) SHG polarimetry curves of PEO–urea and PVA–urea nanofibers, respectively. Reprinted with permission from Ref. [154].

**Table 1 nanomaterials-15-00409-t001:** Ferroelectric nanofibers prepared by electrospinning with ferroelectric granular inclusions. The high-temperature procedure that vaporizes the transporting polymer and coalesces the grains to form ferroelectric fibers is indicated, along with the intended purpose and potential applications. PVP—polyvinylpyrrolidone, PDMS—polydimethylsiloxane, BNKZ—Bi_0.5_(Na_0.82_K_0.18_)_0.5_ZrO_3_, KNNS—(K_0.48_Na_0.52_)(Nb_0.95_Sb_0.05_)O_3_, BZT–BCT—(1 − x)Ba(Zr_0.2_Ti_0.8_)O_3_−x(Ba_0.7_Ca_0.3_)TiO_3_, PAN—polyacrylonitrile.

Polymer	Inclusions	Heat Treatment	Purpose	Ref.
PVP Mw = 1,300,000	KNbO_3_	Heated to 550 °C at a 5 °C/min in air	For humidity sensors	[86]
PVP	(Na,K)NbO_3_	Dried at 100 °C in nitrogen atmosphere for 12 h; annealed at 800 °C for 1 h in air	For biocompatible implants and tissue growth	[87]
PVP	PbZr_1_–_x_Ti_x_O_3_ (PZT) (52/48)	Annealed at 650 °C in air	To measure single-fiber bending piezoelectric voltage	[88]
PVP Mw = 13,000,000	0.96KNNS–0.04BNKZ, BZT–BCT, PZT (52/48)		Energy harvesting	[89]
PVP Mw = 1,300,000	BaTiO_3_ nanofibers in PDMS	Calcined at 1000 °C for 6 h, then inserted in PDMS	Flexible nanogenerators	[90]
PVP Mw = 1,300,000	Aurivillius Bi_5_Ti_3_FeO_15_	Dried in vacuum overnight at 90 °C; calcined at 300 °C for 2 h in oxygen burn PVP; annealed at 600 °C for 2 h in nitrogen; heating rate of 30 °C/h	Multiferroic nanofibers	[91]
PVP Mw = 1,300,000	Co doped Ba_0.7_Sr_0.3_Ti_0.95_Co_0.05_O_3_	Dried at 130 °C for 4 h and heated at 400 °C for 2 h; annealed at target temperatures (700 °C and 600 °C) for 2 h under a heating rate of 2 °C/min	Multiferroic nanofibers	[92]
PVP Mw = 1,300,000	Na_0.5_Bi_0.5_TiO_3_ (NBT) nanofibers in PVDF	Dried at 60 °C for 48 h and then kept at 325 °C and 700 °C for 1 h; afterwards, NBT fibers inserted in PVDF	NBT–PVDF composites in capacitors for energy storage applications	[83]
xylene	Pb(Zr_0.52_Ti_0.48_)O_3_ (PZT)	The as-spun fibers and mats were isochronally sintered in air for two hours at 400, 500, 600, 700, and 800 °C.	To study PZT electrospun nanofiber synthesis	[93]
PVP	Co-doping of Nb^5+^–Nd^3+^ into PZT nanoneedles	Calcination at 800 °C for 2 h	For piezoelectric and high-dielectric-constant applications	[94]
PVP Mw = 1,300,000	vertically aligned ultralong Pb(Zr_0.52_Ti_0.48_)O_3_ (PZT) nanowire	Calcination at 650 °C for 3 h	Wearable energy-harvesting and self-powered devices, flexible fiber nanogenerators	[95,96]
PVP Mw = 1,300,000	BaTiO_3_	Pyrolysis in nitrogen at 900 °C	Photocatalysis	[97]
PVP Mw = 1,300,000	Ba_0.6_Sr_0.4_TiO_3_ (BST)	Different heat treatment temperatures: calcined at 600–800 °C for 2 h in air	To study BST synthesis	[98]
PVP Mw = 50,000	Ba_0.6_Sr_0.4_TiO_3_ (BST)	Dried at 60 °C for 10 h; calcined at 900 °C in air, then included in PVDF to form BST–PVDF composites by drop casting	Nanocomposite capacitors	[99]
PVP	LiNbO_3_ (LNO)	Annealing at 700 °C for 6 h	To study LNO fiber synthesis	[100]
PVP	Mn-doped LiNbO_3_	Annealing at 700 °C	To study Mn–LNO fiber synthesis	[101]
PVP	BaTiO_3_-multiwalled carbon nanotube core–shell fibers	Put in vacuum oven at 70 °C for 1 h, and annealed at 800 °C in a nitrogen atmosphere for 2 h	Flexible piezoelectric pressure sensors	[102]
PAN Mw = 150,000	BiFeO_3_	Calcined at 500 °C for 3 h; then dispersed in PVDF to form a composite	Flexible piezoelectric nanogenerators	[103]
PVP Mw = 1,300,000	BiFeO_3_	Thermal annealing at temperatures from 400 to 600 °C for 2 h in ambient conditions	Photovoltaic devices	[104]
PVP Mw = 1,300,000	BiFeO_3_	Calcined at 520 °C for 2 h in an ambient atmosphere	Photocatalysis	[105]
Nylon-6	BiFeO_3_	Calcined at 600 °C for 2 h in air	Magnetoelectric α_33_ = 0.49 V cm^–1^ Oe^−1^	[106]

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
