# Peer review of "Ferroelectric and Non-Linear Optical Nanofibers by Electrospinning: From Inorganics to Molecular Crystals"

_nanomaterials, 2025, doi:10.3390/nano15050409_

Round 1

Reviewer 1 Report

Comments and Suggestions for Authors

This article presents the ferroelectric and nonlinear optical properties of electrospinning fibers and their potential applications, it is a nice piece of work, but still needs significant improvement. The authors are suggested to address the following comments/suggestions in a revision:

  • The serial numbers in figure 1 are irregularly arranged, and the scale in figure 1c is too close together, the authors are advised to rearrange the layout.
  • Why are there black and gray color fonts in Table 1 and what is the significance?
  • The overall picture in figure 3 is too long, please readjust it, the labeled annealing temperature 1150 ℃ is not consistent with the legend 1050 ℃
  • The data in figure 5 and figure 6 are too homogenous, just giving only SEM maps of the core-shell structure, the authors are suggested to add more data to further confirm this structure.
  • In chapter 3, the authors introduce the core-shell structure and the janus structure, how could this structure improve the ferroelectric properties of fibers? Please add further details.
  • The authors describe the ferroelectric and nonlinear optical properties of the fibers, respectively, so is there any connection between them? Could be improved at the same time? Please add further details.
  • The authors are advised to add practical applications of electrospinning fibers, this is very important.
  • Please keep all pictures in a regular layout with uniform serial numbers.
  • Please revise the article carefully, there are some grammatical errors here.

Comments on the Quality of English Language

There are some grammatical errors in the article, please correct them carefully.

Author Response

Response to Reviewer 1 Comments

1. Summary

Thank you for your thoughtful review and valuable feedback on our manuscript. We have carefully addressed each of your comments, and our detailed responses are provided below. The corresponding revisions and corrections are clearly highlighted/marked using track changes in the re-submitted files for your convenience.

2. Questions for General Evaluation

Reviewer’s Evaluation

Response and Revisions

Is the work a significant contribution to the field?

Is the work well organized and comprehensively described?

Is the work scientifically sound and not misleading?

Are there appropriate and adequate references to related and previous work?  

Is the English used correct and readable? 

3. Point-by-point response to Comments and Suggestions for Authors

This article presents the ferroelectric and nonlinear optical properties of electrospinning fibers and their potential applications, it is a nice piece of work, but still needs significant improvement. The authors are suggested to address the following comments/suggestions in a revision:

Comments 1: The serial numbers in figure 1 are irregularly arranged, and the scale in figure 1c is too close together, the authors are advised to rearrange the layout.

Response 1: Thank you for pointing this out, we have corrected Figure 1

Comments 2: Why are there black and gray color fonts in Table 1 and what is the significance?

Response 2: Thank you for this observation, we have corrected the color fonts in Table 1.

Comments 3: The overall picture in figure 3 is too long, please readjust it, the labeled annealing temperature 1150 ℃ is not consistent with the legend 1050 ℃.

Response 3: Thank you for this comment, we have corrected the inconsistency in the temperature and reformatted the figure.

Comments 4: The data in figure 5 and figure 6 are too homogenous, just giving only SEM maps of the core-shell structure, the authors are suggested to add more data to further confirm this structure.

Response 4: Thank you for pointing this out. We agree with the referee comment. So, we have added to figure 5 images with additional information on the ferroelectric and ferromagnetic properties of core-shell nanofibers. We have also added to figure 6 additional information on the morphological, chemical and magnetoelectric response of composite and Janus fibers.

Comments 5: In chapter 3, the authors introduce the core-shell structure and the janus structure, how could this structure improve the ferroelectric properties of fibers? Please add further details.

Response 5: We haven't found any reports in the literature on the impact of core-shell and Janus structures on the ferroelectric properties of the fibers. However, these structures can enable extra functionalities beyond those of single-phase fibers. Core-shell fibers enable the fabrication of multiferroic materials with increased interfacial interaction between the magnetic and ferroelectric phases, providing new routes to increase in their magnetoelectric response. On the other hand, Janus nanowires or nanofibers, characterized by two distinct phases in their hemi-cylindrical sections, offer the advantage of exposing both phases while maintaining a substantial interfacial contact area. In order to highlight these aspects, we have added the following in section 3:

“Core-shell fibers enable the fabrication of multiferroic systems with increased interfacial interaction between the magnetic and ferroelectric phases, potentially increasing their magnetoelectric response. An interesting example is that reported by Liu et al. [124] observing shifts in the ferromagnetic resonance provoked by an applied electric field. On the other hand, Janus nanofibers, characterized by two distinct phases side-by-side, offer the advantage of exposing both phases while maintaining a substantial interfacial contact area. This can allow for the efficient transfer of mechanical stress transfer between magneto-strictive and piezoelectric components as reported by Mathew et al. [125] who combined magneto-strictive cobalt ferrite with piezoelectric barium titanate to construct a compact sensitive magnetic field sensor.”

Comments 6: The authors describe the ferroelectric and nonlinear optical properties of the fibers, respectively, so is there any connection between them? Could be improved at the same time? Please add further details.

Response 6: Thank you. We would like to clarify that the correlation that exists between ferroelectricity and second harmonic generation in a crystal is the existence of non-centrosymmetry. While this property is sufficient for second harmonic generation to be displayed and therefore the crystal point group is governed by an acentry tensor, a ferroelectric crystal must also belong to a polar point group. As such, every ferroelectric crystal may also generate second harmonic light. However, the reverse is not true, that is a crystal that generates second harmonic generation can only be ferroelectric if its point group is polar. Therefore, the improvement of one property does not imply that the other will also be improved, it might happen for some materials but not for all of them.

Having made this point, there are reports that the anisotropy of second harmonic generation in the well-known ferroelectric crystal Barium Titanate correlates strongly with the polarization orientation of the ferroelectric domains [Wang et al. Scientific Reports 7, 9051 (2017) doi:10.1038/s41598-017-09339-2]. To our knowledge this effect has not been explored in electrospun fibers.

Comments 7: The authors are advised to add practical applications of electrospinning fibers, this is very important.

Response 7: Thank you for encouraging us to make this improvement, we have added the following paragraphs to the Introduction:

“Electrospun polymer fibers embedded with organic molecules have gained significant attention due to their versatility and wide-ranging applications in biomedicine and electronics. In the biomedical domain, these fibers are extensively used for tissue engineering scaffolds and controlled drug delivery systems, as they mimic the extracellular matrix (ECM) and provide a high surface-area-to-volume ratio conducive to cell adhesion, proliferation, and differentiation [33]. These applications benefit from the high surface area to volume ratio and the ease of incorporating a variety of bio-active molecules into the nano-fiber structures. It is possible to endow electrospun nano fibers with diverse physical or chemical properties by incorporating specific tailored nanoparticles [34-36].

In photonics, electrospun fibers exhibit promising nonlinear optical properties and enhanced light emission characteristics. As will be explored in this review, notable second harmonic generation (SHG) has been observed in several polymer-organic and semi-organic electrospun systems, underscoring their potential for frequency-doubling applications. Furthermore, conjugated polymer fibers have demonstrated enhanced polarized photoemission coupled with self-waveguiding behavior [37,38], which is particularly advantageous for nanophotonic devices and sensors in the blue spectral region, an area where solid-state photonic sources are challenging to develop. These op-tical functionalities are attributed to the alignment of the molecular dipoles by the strong electric fields employed in the electrospinning process and the smooth morphology and flexibility of the fibers, which support efficient waveguiding and light manipulation.

Incorporating ferroelectric organic and semi-organic nanocrystals into polymeric fibers through electrospinning has led to the development of sensors and energy harvesting systems. This approach can potentially harness both the pyroelectric and piezoelectric effects, synergistically combining the functional properties of nanocrystals with the flexibility and high surface area of electrospun fibers. Applications of these nanofiber-based ferroelectric systems span a broad spectrum of applications, with particular emphasis on energy harvesting technologies. Several recent reviews focus on the applications of ferroelectric composites and piezoelectric electrospun fibers [11,39-47]. However there remains a notable gap in the literature concerning the fundamental physical properties of electrospun single-phase inorganic ferroelectric nanofibers and organic ferroelectric nanofibers beyond polyvinylidene fluoride (PVDF).”

Comments 8: Please keep all pictures in a regular layout with uniform serial numbers.

Response 8: Thank you for your feedback. We have checked that all figures are numbered sequentially with the figure legends uniformly formatted.

Comments 9: Please revise the article carefully, there are some grammatical errors here.

Response 9: Thank you for this alert. We have corrected all the grammatical errors we found.

Reviewer 2 Report

Comments and Suggestions for Authors

The manuscript is well written and well organized. For completeness, I suggested that some points be added that I consider important for this type of review.

In item 3.3 there could be more figures showing composite multiferroic nanowires.

There are nanowires sintered by electrospinning that use single-phase multiferroic materials. These materials have some advantages over composites. This type of material could be mentioned.

Author Response

Comments 1: In item 3.3 there could be more figures showing composite multiferroic nanowires.

Response 1: We thank the reviewer for their suggestion to include more figures illustrating composite multiferroic nanowires. In response, we have expanded Figures 5 and 6 to provide additional details on the structural and functional characteristics of the fibers:

  • In Figure 5, we have incorporated new images highlighting the ferroelectric and ferromagnetic properties of core–shell nanofibers, offering a more comprehensive view of their multifunctional behavior.
  • In Figure 6, we have added further data on the morphological features, chemical composition, and magnetoelectric response of both composite and Janus fibers, seeking to provide a deeper view of their structure–property relationships.

We believe these additions address the reviewer’s concern and contribute to a clearer and more complete presentation.

Comments 2: There are nanowires sintered by electrospinning that use single-phase multiferroic materials.

Response 2: We thank the reviewer for their observation. We acknowledge that single-phase multiferroic nanowires have been reported in the literature (e.g., npj Computational Materials, 10, 187 (2024); Scripta Materialia, 89, 17–20 (2014); Chemical Physics Letters, 579, 78–84 (2013), among others). However, these studies fall outside the scope of our review, as they do not involve nanowire fabrication via electrospinning.

To the best of our knowledge, only BiFeO₃ (BFO) nanofibers have been produced by electrospinning (Phys. Status Solidi RRL, 6, No. 6, 244–246 (2012)). To clarify this point, we have added the following sentence to Section 3.3 and included a reference in Table 1 of the manuscript:

“Nevertheless, single-phase BiFeO₃ electrospun nanofibers have been reported [103-106], exhibiting a high magnetoelectric coupling coefficient (α33) comparable to that of BFO bulk and thin films. However, research on single-phase multiferroic electrospun nanofibers remains limited, highlighting the need for the development and characterization of other materials.”

We believe this addition addresses the reviewer’s concern while maintaining the focus of our review on electrospun multiferroic nanofibers.

Comments 3: These materials have some advantages over composites. This type of material could be mentioned.

Response 3: We agree that these materials are significant, particularly due to their intrinsic magnetoelectric coupling. However, as discussed in our previous response, only a limited number of single-phase multiferroic systems produced by electrospinning have been reported to date.

To address this point, we have added information on the known electrospun single-phase systems to the manuscript for clarity. Specifically, we highlighted that BiFeO₃ (BFO) nanofibers, the most notable example, exhibit a high magnetoelectric coupling coefficient (α33), comparable to that of bulk and thin-film counterparts.

This addition, along with the references included in Table 1 and Section 3.3, underscores the potential of single-phase multiferroic nanowires and the need for further research to expand the range of materials produced via electrospinning.

Reviewer 3 Report

Comments and Suggestions for Authors

The paper concentrates on a specific area within nanomaterials research – the use of electrospinning to create functional nanofibers with ferroelectric and nonlinear optical properties. This focused approach allows for a deeper dive into the topic than a broader review might offer. The work is generally well-organized, with clear section headings and subheadings. The figures and tables are helpful in summarizing key information, though some could be improved for clarity (more concise labels, better figure resolution). The review is generally scientifically sound, citing appropriate references to support claims.

Reading the manuscript, the following issues can be found:

- While the authors mention limitations, a more critical evaluation of the strengths and weaknesses of different electrospinning parameters and processing techniques would enhance the scientific rigor.

- The depth of the analysis varies across different material systems. Some materials are discussed in greater detail than others.

- Claims, particularly regarding performance metrics in specific applications, could benefit from more detailed supporting evidence.

- The emphasis is heavily weighted toward synthesis techniques and material properties. While applications are mentioned, a more thorough exploration of the practical applications of these nanofibers and their real-world performance would enhance the review. For instance, discussing challenges in scaling up production for commercial applications would add significant value.

- The review almost entirely ignores producing these nanofibers' economic and environmental aspects. A discussion of the costs associated with various materials and techniques, as well as potential environmental impacts, would increase the review's completeness.

- For biomedical applications, the toxicity and biocompatibility of these materials should be considered. This is missing from the review.

- The long-term stability of the nanofibers under various conditions (temperature, humidity, etc.) should be discussed, as this is crucial for practical applications.

Addressing these disadvantages and filling the gaps would significantly enhance the quality and impact of this review article, transforming it from a descriptive overview to a more critical and insightful analysis of the field.

Author Response

3. Point-by-point response to Comments and Suggestions for Authors

The paper concentrates on a specific area within nanomaterials research – the use of electrospinning to create functional nanofibers with ferroelectric and nonlinear optical properties. This focused approach allows for a deeper dive into the topic than a broader review might offer. The work is generally well-organized, with clear section headings and subheadings. The figures and tables are helpful in summarizing key information, though some could be improved for clarity (more concise labels, better figure resolution). The review is generally scientifically sound, citing appropriate references to support claims.

Reading the manuscript, the following issues can be found:

Comments 1: While the authors mention limitations, a more critical evaluation of the strengths and weaknesses of different electrospinning parameters and processing techniques would enhance the scientific rigor.

Response 1: While we agree that a detailed discussion of electrospinning parameters and processing techniques is important, the primary aim of this review is to highlight materials whose properties are enhanced when processed as nanofibers via electrospinning, rather than to provide a comprehensive analysis of the electrospinning process itself.

We note that different materials require specific adjustments to the numerous electrospinning parameters, such as solution properties, electric field strength, and collector configuration. However, these aspects are typically discussed in detail within the original publications cited in our references, as they are closely tied to the material systems under study. Nevertheless, we feel that in part we have addressed this concern in section 2.2. highlighting how key electrospinning parameters can influence nanofiber morphology and functional properties.

Comments 2: The depth of the analysis varies across different material systems. Some materials are discussed in greater detail than others.

Response 2: We thank the reviewer for their observation. We acknowledge that the depth of analysis varies across the different material systems discussed. This discrepancy primarily reflects the uneven availability of published data for each system. Nevertheless, we have made an effort to provide a comprehensive analysis based on the current state of the literature. Importantly, we hope that our review will inspire further investigation into less-explored material systems in future studies.

Comments 3: Claims, particularly regarding performance metrics in specific applications, could benefit from more detailed supporting evidence.

Response 3:  We thank the reviewer for this valuable observation. Our primary objective in this review is to highlight recent advances in enhancing ferroelectricity and nonlinear optical responses in organic materials through their incorporation into polymeric fibers via electrospinning. In this context, the key performance metrics are the piezoelectric constant and the effective second-order susceptibility, which we have discussed throughout the text.

Comments 4: The emphasis is heavily weighted toward synthesis techniques and material properties. While applications are mentioned, a more thorough exploration of the practical applications of these nanofibers and their real-world performance would enhance the review. For instance, discussing challenges in scaling up production for commercial applications would add significant value.

Response 4: We acknowledge that our review primarily emphasizes material properties, as our chief intention is to highlight the enhancement of properties in a specific class of hybrid materials—obtained from polymeric solutions doped with semi-organic and all-organic molecular crystals exhibiting second harmonic generation or ferroelectricity—fabricated using the electrospinning technique. While we agree that practical applications and scaling-up challenges are important, we have not made these the focus of our review because large-scale manufacturing processes for these hybrid systems remain largely unexplored in the current state of the art. Instead, our aim is to emphasize the enhanced properties exhibited by these systems, which we believe can drive future efforts toward their practical implementation and commercialization. To clarify our stance, we have added text to the Introduction highlighting the potential of these hybrid materials for future technological and commercial developments.

Comments 5: The review almost entirely ignores producing these nanofibers' economic and environmental aspects. A discussion of the costs associated with various materials and techniques, as well as potential environmental impacts, would increase the review's completeness.

Response 5: While we agree that the economic and environmental aspects of electrospun nanofiber production are important topics, they fall beyond the intended scope of our review. As discussed in our previous responses, our primary focus is on the enhancement of functional properties in hybrid materials produced via electrospinning, rather than on the technological, economic, or environmental implications of the process. We believe that a thorough analysis of economic feasibility and environmental impact would be more appropriate for a dedicated review focused on these aspects.

Comments 6: For biomedical applications, the toxicity and biocompatibility of these materials should be considered. This is missing from the review.

Response 6: As noted in our previous responses, the primary focus of this review is to present the ferroelectric, piezoelectric, and nonlinear optical properties of hybrid materials produced via electrospinning, rather than to explore specific applications. While we agree that toxicity and biocompatibility are critical considerations for biomedical applications, these topics fall outside the intended scope of our review. A thorough discussion of these aspects would be more appropriate for a dedicated review focusing on biomedical applications of electrospun nanofibers.

Comments 7: The long-term stability of the nanofibers under various conditions (temperature, humidity, etc.) should be discussed, as this is crucial for practical applications.

Response 7: We thank the reviewer for raising this important point. We agree that long-term stability under various conditions, such as temperature and humidity, is critical for practical applications. However, because our review covers a broad range of materials, including both inorganic and organic systems, it is challenging to generalize stability trends across all systems. In general, we note that embedding functional materials within polymeric fibers via electrospinning often provides additional protection against oxidation and environmental degradation, enhancing their long-term stability. For more detailed information on the stability of specific material systems, we hope that the interested reader will consult the original studies cited in our review, where these aspects are frequently discussed in the context of individual materials.

Round 2

Reviewer 1 Report

Comments and Suggestions for Authors

Accept in present form

Author Response

We sincerely appreciate your comments and suggestions, which have helped us improve our review. Thank you for your time and valuable insights.

Reviewer 2 Report

Comments and Suggestions for Authors

Dear authors. Thank you for the comments. I'm satisfied with the report.

Author Response

Thank you for your comments. We appreciate your feedback and are glad that our revisions have addressed your concerns.

Reviewer 3 Report

Comments and Suggestions for Authors

Dear Authors,

I see that you don't want to change significantly your initial version of the manuscript and for any questions raised about the topic your justification is "out of the scope". You have to take into account that all of these issues are in the scope of any material - long-term stability, environmental aspect, scalability for future applications. Otherwise, why do we investigate them? The review papers are not only a mix of someone else's results but also your own analyses, perspectives, and discussion about the expectations, what is possible and what is not possible. Only in this way, you may truly identify significant gaps in the research and suggestions of where research might go next.

Author Response

"Dear Authors,

I see that you don't want to change significantly your initial version of the manuscript and for any questions raised about the topic your justification is "out of the scope". You have to take into account that all of these issues are in the scope of any material - long-term stability, environmental aspect, scalability for future applications. Otherwise, why do we investigate them? The review papers are not only a mix of someone else's results but also your own analyses, perspectives, and discussion about the expectations, what is possible and what is not possible. Only in this way, you may truly identify significant gaps in the research and suggestions of where research might go next."

Authors response:

In order to address the expressed concerns, a new section has been incorporated. This section is entitled "Perspectives for Applications: Stability, Biocompatibility, Scalability, and Environmental Impact." Within this section, the aforementioned aspects are discussed in detail.

  1. Perspectives for Applications: Stability, Biocompatibility, Scalability, and Environmental Impact

Recent advances in the development of electrospun ferroelectric and nonlinear optical nanofibers have demonstrated their potential for a wide range of applications, including electronics, photonics, sensors, and biomedical devices. It has been forecast-ed that the global market for polymeric nanofibers should grow from $2.2 billion in 2021 to $6.7 billion by 2026 with a compound annual growth rate of 25.1% for the peri-od 2021–2026 [171,172]. However, for these materials to continue to improve their transition from laboratory research to commercial applications, several critical aspects must be considered, including their long-term stability, biocompatibility, large-scale production feasibility, and environmental impact.

6.1. Long-term Stability

The long-term performance of electrospun nanofibers is an important factor in determining their viability for industrial and biomedical applications. Their stability can be affected by humidity, temperature fluctuations, and mechanical stress. In the case of inorganic ferroelectric nanofibers, degradation mechanisms such oxygen content can compromise functionality over time. As inorganic nanofibers tend to be brittle, they have been successfully included inside polymers which gives them flexibility, strength and long-term stability [99,103,105,109]. On the other hand, organic-based ferroelectric and non-linear optical nanofibers exhibit stable molecular organization, as they do not undergo rearrangements that could diminish performance. Organic inclusions, when embedded into electrospun nanofibers, maintain their properties for more than one year, as the polymer matrix acts as a shielding by inhibiting oxidation [173-175]. Additionally, when non-centrosymmetric organic molecules crystallize within nano- and microfibers, they maintain their structural integrity, mitigating the risk of molecular reorganization. The electrospinning process effectively preserves molecular orientation, ensuring that organic molecules form single crystal-like structures that keep their optical and piezoelectric properties and enhance their non-linear optical properties [21] which are important for photonic applications.

6.2. Biocompatibility

Biocompatibility is a key consideration for applications in biomedical engineering, including biosensors, tissue scaffolds, and implantable devices. Certain ferroelectric polymers, such as polyvinylidene fluoride (PVDF) and piezoelectric polymers like poly-L-lactic acid (PLLA) have demonstrated biocompatibility [176,177], making them suitable for use in biomedical applications. By embedding into those polymers non-toxic all organic molecules such as urea and β-glycine as well semi-organic lead-free perovskites retaining their nonlinear optical, piezoelectric and pyroelectric properties, it is possible to manufacture disposable, environmentally safe active electrospun fiber mats to be integrated into appropriate devices. On the other hand, inorganic ferroelec-tric nanofibers may require additional surface modifications to minimize cytotoxicity. Additionally, lead-free ferroelectric compounds are favoured due to their reduced toxici-ty. In this respect, important ferroelectrics, such as lead-free BaTiO3, have shown very good biocompatibility, even having been used to coat other more toxic compounds, in the context of their use in biomedical applications [178]. Ferrites, such as CoFe2O4 have also shown good biocompatibility, particularly when embedded inside the polymeric fibers [179,180]. Further in vitro and in vivo studies are necessary to evaluate long-term interactions of these nanofibers with biological tissues.

6.3. Scalability of Production

For practical implementation, the scalable production of electrospun nanofibers with consistent properties is essential. Electrospinning offers significant advantages in terms of tunability and ease of processing. However, challenges remain in achieving uniform fiber morphology and composition when moving from laboratory-scale to industrial-scale production [181]. High-throughput electrospinning techniques, such as multi-jet, could provide solutions for scaling up manufacturing while maintaining the functional properties of the nanofibers [181]. Notable examples of multiple-needle electrospinning include the Nanospinner developed by Inovenso Inc., which involves 110 needles and is used the commercial production of filtration membranes and medical devices [182]. Needleless electrospinning has also found commercial use for high throughput nanofiber production, such as the Nanospider Production Line developed by Elmarco Inc., [183] which uses a wire electrode to eject multiple jets. Nevertheless, the reproducibility of fiber alignment, deposition, and integration into devices must continue to be optimized, for commercialization.

6.4. Environmental Impact

As sustainability becomes a growing concern, the environmental impact of nanofiber production and disposal must be assessed [171]. Some ferroelectric and nonlinear optical nanofibers contain lead-based materials or organic molecules (some examples are included in tables 1 to 3), which raise concerns regarding toxicity and recyclability [184]. Developing lead-free alternatives, such as lead-free barium titanate-based ferroelectrics [185,186], semi-organic perovskites and organic molecules based on amino acids and peptides can provide more environmentally friendly solutions [184,187]. Some of these solutions were already demonstrated on the present review. The incorporation of functionalized nano and microfibers mats into appropriate devices contributes to materials resources sustainability. This is because the embedded functional materials are under the form of powders, which are dissolved into a polymer matrix. Therefore, there is no need for using bulk materials, which are expensive to produce and involve extra resources. Additionally, the selection of biodegradable polymers and green solvents for electrospinning can reduce the ecological footprint of nanofiber production. Future research should focus on life cycle assessments and end-of-life strategies to ensure that these materials align with sustainable development goals.

Round 3

Reviewer 3 Report

Comments and Suggestions for Authors

I have no comments.